

# Accounting for surface waves improves gas flux estimation at high wind speed in a large lake

Pascal Perolo[1], Bieito Fernandez Castro[2,3], Nicolas Escoffier[1], Thibault Lambert[1], Damien Bouffard[4], Marie-Elodie Perga[1]

[1]Institute of Earth Surface Dynamics, University of Lausanne, Lausanne, 1015, Switzerland
[2]Physics of Aquatic Systems Laboratory, Margareth Kamprad Chair, Swiss Federal Institute of Technology Lausanne, Lausanne, 1015, Switzerland
[3]Ocean and Earth Science, University of Southampton, National Oceanography Centre, Southampton, SO14 3ZH, United Kingdom
[4]Eawag, Swiss Federal Institute of Aquatic Science and Technology, Surface Waters – Research Management, Kastanienbaum, 6047, Switzerland

*Correspondence to*: Pascal Perolo (pascal.perolo@unil.ch)

**Abstract.** The gas transfer velocity ($k$) is a major source of uncertainty when assessing the magnitude of lake gas exchange with the atmosphere. For the diversity of existing empirical and process-based $k$ models, the transfer velocity increases with the level of turbulence near the air-water interface. However, predictions for $k$ can vary by a factor of 2 among different models. Near-surface turbulence results from the action of wind shear, surface waves and buoyancy-driven convection. Wind shear has long been identified as a key driver, while recent lake studies have shifted the focus towards the role of convection, particularly in small lakes. In large lakes, wind fetch can however be long enough to generate surface waves and contribute to enhance gas transfer, as widely recognised in oceanographic studies. Here, field values for gas transfer velocity were computed in a large hardwater lake, Lake Geneva, from $CO_2$ fluxes measured with an automated (forced diffusion) flux chamber and $CO_2$ partial pressure measured with high frequency sensors. $k$ estimates were compared to a set of reference limnological and oceanic $k$ models. Our analysis reveals that accounting for surface waves generated during windy events significantly improves the accuracy of $k$ estimates in this large lake. The improved $k$ model is then used to compute $k$ over a one-year time-period. Results show that episodic extreme events with surface waves (6 % occurrence, significant wave height > 0.4 m) can generate more than 20 % of annual cumulative $k$ and more than 25 % of annual net $CO_2$ fluxes in Lake Geneva. We conclude that for lakes whose fetch can exceed 15 km, $k$-models need to integrate the effect of surface waves.

## 1 Introduction

Lakes are universally regarded as significant sources of $CO_2$ to the atmosphere, however, the accurate quantification of the magnitude of such emissions remains to date challenging (Cole et al., 2007; Tranvik et al., 2009; Raymond et al., 2013). While $CO_2$ fluxes can be directly measured with floating chamber or eddy covariance systems (Vachon et al., 2010; Vesala et al., 2006), these approaches suffer from limited time and space integration (from minutes to hours, and centimetres to metres;



Klaus and Vachon, 2020). Long-term direct flux measurements are thereby mostly restricted to small lakes (Huotari et al., 2011) and fluxes remain mostly estimated with models. $CO_2$ fluxes at the surface of lakes operate through a net diffusive transport, therefore obeying the Fick's first law:

$$F = k\alpha\Delta pCO_2 , \hspace{5cm} (1)$$

where $F$ (mol m$^{-2}$ s$^{-1}$ but often expressed as µmol cm$^{-2}$ h$^{-1}$) is the $CO_2$ gas flux, $\alpha$ is the $CO_2$ solubility coefficient (µmol cm$^{-3}$ µatm$^{-1}$), $\Delta pCO_2$ is the gradient of partial pressure of $CO_2$ ($pCO_2$) between the water and the atmosphere corrected for altitude (µatm); and $k$ is the gas transfer velocity (cm h$^{-1}$).

Lake carbon emissions are therefore primarily driven by the gradient of partial pressure of $CO_2$ between the surface lake water and the atmosphere, but the gas-transfer velocity controls the rate of $CO_2$ exchange across the lake-atmosphere interface. Assessing the amount of lake $CO_2$ emissions to the atmosphere has been a major issue, starting with the Cole and al's 1998 seminal paper, with debates regarding both the representativeness of the measurements and the optimal conceptual model for air-water gas transfer (e.g., MacIntyre et al., 2001; Borges et al., 2004). As recent developments in sensor technologies allow continuous and accurate measurements of aqueous $CO_2$ concentrations, the gas transfer velocity remains, to date, the main source of uncertainties, which hinders attempts to achieve full carbon budgets (Dugan et al., 2016) or to quantify greenhouse gas emissions by lakes, at local, regional, or worldwide scales (Maberly et al., 2012; Raymond et al., 2013; Engel et al., 2018).

$k$ is inherently tied to turbulent mixing within the surface boundary layer, which enhances the diffusive gas exchange by renewing the surface mass content (Zappa et al., 2007). At the lake-atmosphere interface, turbulent mixing is the product of wind shear, buoyancy flux, and wind-driven surface waves, whose effect can be split into wave action and wave breaking, the latter producing air bubble and water spray (Fig. 1; Wüest and Lorke, 2003, Soloviev et al., 2007). Regarding the prominent role of wind action on surface turbulence, first quantitative models have empirically scaled $k$ to wind-speed (referenced at a 10 m height; $U_{10}$), as a proxy for the level of wind-driven turbulence (Fig. 1; Cole and Caraco, 1998; Crusius and Wanninkhof, 2003). The parameterizations of the $k$-wind relationships vary between authors (e.g., Klaus and Vachon 2020), as a likely consequence of the local characteristics of the lakes used in the calibration datasets (Table 1). Yet, all studies suggested a polynomial relationship between $U_{10}$ and $k$ with order larger than unity. Further development of empirical models integrated the lake surface area as a second parameter in the $k$-wind relationships, to account for the role of the fetch length for wind action (Vachon and Prairie, 2013). Generally, empirical wind-based models tend to underestimate fluxes, especially at low-wind speed (e.g., Schubert et al., 2012; Heiskanen et al., 2014; Mammarella et al., 2015) where turbulent mixing through buoyancy flux is expected to take over wind shear. Besides, these empirical models require a proper calibration each time they are applied in a new system with different characteristics, i.e., a new set of lakes and/or meteorological conditions (Klaus and Vachon, 2020), hence limiting their universal applicability.

**Figure 1**





In parallel to empirical wind-based models, process-based models attempt to link $k$ directly to near-surface turbulence. The surface renewal model (SRM) is one of the first, and still most widely used, theories (Danckwerts 1951; Lamont and Scott, 1970) with $k$ depending on the product of the turbulent kinetic energy dissipation rate ($\varepsilon$) and the kinematic viscosity of water ($v$), both to a power of one quarter as follows:

$$k = a_1 (\varepsilon v)^{1/4} Sc^{-1/2} ,$$ (2)

with $a_1$ a calibration constant parameter and $Sc$, the Schmidt number. Recently Lorke and Peeters (2006) and Katul and Liu (2017) demonstrated that this relationship, to which different approaches converge, can be seen as a universal scaling. As opposed to the practical empirical models presented above, process-based models have the potential to predict $k$ using the turbulent dissipation rate over a wide range of environmental conditions extending beyond those encountered in the calibration dataset (Zappa et al., 2007). As for lakes, SRM $k$-models have so far considered the friction velocity at the water side ($u_{wat}$)

and the turbulence created by thermal convection using the buoyancy flux at the surface ($B_0$) (Fig. 1; Eugster et al., 2003; MacIntyre et al., 2010; Read et al., 2012; Tedford et al., 2014; Heiskanen et al., 2014). Noteworthily, the SRM approach leads to $k$ being related to $u_{wat}$ (or $U_{10}$) to the first order (Wanninkhof, 1992; Lorke and Peeters 2006; see also Material and Methods section), while empirical models described above predict a higher order polynomial relationship. This inconsistency is tentatively solved in oceanography by adding another source of gas exchange associated with wind-waves whitecaps. Early

gas flux parameterization already accounted for wind and buoyancy-driven turbulence together with surface waves (Fig. 1; Woolf et al., 1997; Soloviev et al., 2007; Fairall et al., 2011). Yet, the buoyancy-driven contribution can often be neglected in oceanography, and recent efforts have been dedicated to a better parameterization of the bubble enhancement term (Fig. 1; Deike and Melville, 2018). In lakes, wind fetch can be long enough to generate surface waves (Wanninkhof, 1992; Frost and Upstill-Goddard, 2002; Borges et al., 2004; Guérin et al., 2007), thus implying that surface waves could be a significant driver

of $k$ and subsequent $CO_2$ fluxes (Schilder et al., 2013; Vachon and Prairie, 2013). Insofar, the role of surface waves has been essentially accounted empirically in lake $k$-models, through the polynomial scaling to $U_{10}$ in wind-based models, and most often neglected in studies using process-based parameterizations (mainly SRM) (e.g., Read et al., 2012). While this approximation may be appropriate for small-shielded lakes, it is likely to be insufficient in larger, long-fetched lakes.

Herein, we aim to identify the most adequate $k$-model for Lake Geneva, a large clear hardwater lake in the Swiss Alps, to assess $k$ values over a full annual cycle. We compare the performances of different models of gas transfer velocity, in their original or slightly modified published formulations from the limnological and oceanic literatures. This set of models includes different levels of complexity, ranging from empirical models integrating wind speed and lake size, to process-based models including wind shear, convection, and surface waves. Continuous $\Delta pCO_2$ measurements by in situ automated sensors and $CO_2$

fluxes, obtained from a new generation of automated (forced diffusion) flux chamber, were collected during specific periods of intensive field survey covering a wide range of natural conditions. Empirical $k$ values computed from chamber data are then compared to outputs from the different $k$-models. Owing to the size of Lake Geneva, we anticipate that models accounting,





implicitly or explicitly, for the four key exchange drivers (i.e., wind shear, convective mixing, wave action and bubble formation) will show the highest accuracy and precision in their estimation of $k$ and that a precise integration of surface wave effects in such a large system should enhance model predictions. Thereafter, the relative distribution of these components is computed over a full year and analysed in the scope of the temporal variability of the gas transfer velocity. Finally, we expect that extreme wind and associated wave events should contribute disproportionately to accumulated $k$ values over the year. In such a case, episodic weather events could generate large $CO_2$ fluxes over very short timescales that should be accounted for when computing annual $CO_2$ emission budgets.

## 2 Material and Methods

### 2.1 Study Site

Lake Geneva is a peri-alpine lake defining part of the Swiss-French border, at 372 m above sea level (46° 26' N; 6° 33' E). Its surface area (582 km$^2$) and its maximum depth (309 m) make it the largest freshwater body in Western Europe, with a volume of 89 km$^3$ (Fig. 2). Lake Geneva is monomictic. The two prevailing winds are nearly opposed and come from southwest and northeast respectively (Fig. 2). The lake water has been monthly or fortnightly surveyed from the late 1950's (OLA-IS, AnaEE-France, INRAE of Thonon-les-Bains, CIPEL, Rimet et al., 2020). Surface $CO_2$ concentrations, as computed from the routine temperature, alkalinity, and pH measurements (Stumm and Morgan, 1981), show a typical seasonal cycle with high, supersaturated values during winter mixing and values below saturation in summer (Perga et al., 2016).
**Figure 2**

### 2.2 Field data at LéXPLORE

All field data were collected from the LéXPLORE platform, a 10 m by 10 m pontoon equipped with high-tech instrumentation and installed on Lake Geneva in 2019. LéXPLORE is moored at a 110 m depth, 570 m off the northern lake shore (Fig. 2).

On LéXPLORE, local weather conditions (air temperature, wind speed and direction, relative humidity, short wave radiation and atmospheric pressure) were continuously recorded (10 minutes intervals) by a Campbell Scientific Automatic Weather Station. Lake surface temperature was measured every minute at 50 cm-depth using a Minilog II-T (Vemco, resolution 0.01° C). Partial pressure of water surface $CO_2$ (pCO$_2$) was also measured at 50 cm-depth during specific surveys (see flux measurement) using a miniCO$_2$ sensor (Pro-Oceanus Systems Inc.) with an accuracy of ± 30 ppm. Values of pCO$_2$ in ppm were converted into µatm following the basic equation correcting for altitude (Russell and Denn, 1972). We therefore assume that the concentration and the temperature are homogeneous over the first 50 centimetres.

Fetch distance (m) from LéXPLORE to the lake shores considering wind direction was computed using data from the Federal Office of Topography online portal (Swisstopo-geoportal: geo.admin.ch). The position of LéXPLORE is particularly relevant





for this study as the fetch ranges from ~0.5 km to ~30 km for the two prevailing winds. Significant wave height, $H_s$, (in m)

was computed after Hasselmann et al. (1973) according to:

$$H_s = 1.6 \cdot 10^{-3} \cdot U_{10} \cdot (Fetch/g)^{1/2} \, , \tag{3}$$

where $g$ is the gravitational constant. This equation is equivalent to the formulation by Carter (1982) that is more widely used in the oceanic literature. Simon (1997) tested the model for significant wave heights in Lake Neuchâtel (a nearby but smaller lake than Lake Geneva) with a fetch distance of 9 km. His results showed that, beyond a critical threshold of wind value (~5

m s$^{-1}$) wave breaking occurs faster and with a higher probability in the case of not fully developed surface waves. Such waves are characterized by steeper slopes that favour their breaking (Wüest and Lorke, 2003).

The net $CO_2$ flux at the lake-atmosphere interface, $F$, was directly measured with an automated (forced diffusion) floating $CO_2$ flux chamber (eosFD, eosense: environmental gas monitoring; Risk et al., 2011) originally developed for soil flux studies. The

flux chamber had a detection limit close to 0.05 μmol cm$^{-2}$ h$^{-1}$ and measured $F$ every 15-minutes in summer and 30-minutes in winter for battery-saving purpose. The standard floating chambers require quiet surface conditions (e.g., Cole et al., 2010; Vachon et al., 2010; Bastviken et al., 2015), thus limiting studies from low to moderate wind speed conditions. One typical problem with floating chambers arises from the possible atmospheric leakage under rough surface. Our flux chamber was specifically conceived to increase stability under windy conditions (Fig. A1). We tested the performance of the floating

chamber by comparing the standard deviation of the $CO_2$ concentrations of the atmosphere and in the chamber estimated from two separated cavities (Fig. A1; Risk et al., 2011). We did not observe any difference in the standard deviation between high and low wind conditions (Fig A3), suggesting that the measured fluxes remained reliable at high wind speed.

We assessed the performances of our flux chamber during 5 specific periods over the annual cycle (i.e., 13$^{th}$–14$^{th}$ June 2019,

27$^{th}$–28$^{th}$ August 2019, 1$^{st}$–5$^{th}$ October 2019, 18$^{th}$–20$^{th}$ December 2019, and 20$^{th}$–26$^{th}$ February 2020). To select the most robust dataset to compare with $k$ estimates derived from models, we discarded flux data that were below the detection limit, as well as $CO_2$ gradients that ranged within the uncertainty of sensors (i.e., $\pm$ 20 ppm for air and $\pm$ 30 ppm for water leading to $\pm$ 50 ppm) (Fig. A2). Accordingly, we were able to retain the most robust data points during the following deployment periods: 18$^{th}$–20$^{th}$ December 2019 and 20$^{th}$–26$^{th}$ February 2020. Finally, all these field data were standardized at a 1-hour timestep.

**2.3 Computed $k$ values from field data**

$k$ values (cm h$^{-1}$) from field observations ($k_{obs}$) were computed from the gas transfer velocity equation:

$$k_{obs} = F/(\alpha \cdot \Delta pCO_2), \tag{4}$$

where $F$ is the measured $CO_2$ flux (μmol cm$^{-2}$ h$^{-1}$), $\alpha$ is the gas solubility coefficient (μmol cm$^{-3}$ μatm$^{-1}$), which depends on the measured water temperature (Wanninkhof, 1992). $\Delta pCO_2$ is the differential of pCO$_2$ measured at 0.5 m below the surface





(pCO$_2^{water}$) and pCO$_2$ at saturation (pCO$_2^{sat}$) in ppm measured from the flux chamber corrected by altitude (µatm). $k_{obs}$ was

then standardized in $k_{600}$ using the dimensionless Schmidt number ($Sc$) of CO$_2$ by the equation: $k_{600-obs} = k_{obs} \cdot (600/Sc)^{-1/2}$ (600 for freshwater standardized at 20° C).

**2.4 Models for air–water gas transfer velocity**

After years of debate, a consensus begins to emerge on the relationship linking $k$, intensity of turbulence and $Sc$ (Eq. 2), even

when starting from different physical assumptions (see Katul et al., 2018). In this study, we selected six parameterizations widely used in limnology and oceanography combining specific calibration characteristics (Table 1). We first show that they can all be expressed following Eq. (2) for wind shear and convection, despite their different formulations. Then, we develop the effects of surface waves from oceanic models and adapt the wave action for a large lake. The final lake model integrating wave effect is ultimately calibrated using our field data (Table 1).

**Table 1**

**2.4.1 Wind shear**

We start with the case where near-surface dynamics are driven by a weak to moderate wind, in absence of heat exchange. In this case, the contribution of surface waves can be neglected and the wind stress ($\tau_0 = \rho_{air} \cdot C_{10} \cdot U_{10}$) is equal to the tangential shear stress ($\tau_t = \rho_{wat} \cdot u_{*,wat}^2$). The relationship between $\varepsilon$ and the sheared velocity on the water side, $u_{*,wat}$, is then derived

from a law-of-the-wall scaling for the velocity profile: $\varepsilon = u_{*,wat}^3 / \kappa z(0)$ with $K$ being the von Kármán constant (= 0.41) and $z(0)$ the thickness of the diffusive boundary layer. This relationship leads to:

$$k_{NB} = a_1 \cdot \left( v \, u_{*,wat}^3 / \kappa z(0) \right)^{1/4} Sc^{-0.5} , \qquad (5)$$

The challenge is then to define $z(0)$. Tedford et al. (2014) followed an ad hoc observational approach and chose $z(0)= 0.15$ m, as the shallower depth where $\varepsilon$ was measured. In contrast, theoretical studies linked $z(0)$ to the thickness of the diffusive

or viscous sublayer (~0.1–1 cm). In line with theory, we scale this layer as $z(0) = cv/u_{*,wat}$ (Wüest and Lorke, 2003; Lorke and Peeters, 2006) with $c$ as a constant value. Taking $c = 114$ (Soloviev et al., 2007), the thickness of this layer typically ranges from 0.04 to 0.14 m under a wind regime of 10 to 1 m s$^{-1}$. We therefore modify Eq. (5) as to compute the interfacial (no-bubble, NB) exchange coefficient:

$$k_{NB} = a_1 u_{*,wat} (1/\kappa c)^{1/4} Sc^{-0.5}, \qquad (6a)$$

or

$$k_{NB} = a_1 (\rho_{air}/\rho_{wat}) C_{10} U_{10} (1/\kappa c)^{1/4} Sc^{-0.5} , \qquad (6b)$$

These equations show that the SRM formulation (Soloviev et al., 2007; Read et al., 2012; Table 1 and Fig. B1) is analogous to the COAREG flux algorithm (Fairall et al., 2011), and the formulation used in Deike and Melville (2018) with the sheared





velocity on the atmosphere side, $u_{*,atm}$ (Table 1: *DM18*). Indeed, when equating the expression by Deike and Melville (2018):

$\quad k_{NB} = A_{NB} u_{*,atm} (Sc/600)^{-1/2} = A_{NB} u_{*,wat} (\rho_{wat}/\rho_{atm})^{1/2} (Sc/600)^{-1/2}$ , $\qquad$ (7)

with (6a), we find that the coefficient $a_1 = 0.29$ in Soloviev et al., (2007) and Read et al., (2012) is essentially equivalent to the coefficient $A_{NB} = a_1 (\kappa c)^{-1/4} (1/600)^{1/2} (\rho_{wat}/\rho_{air})^{-1/2} \approx 1.5 \times 10^{-4}$ in Deike and Melville (2018) (Fig. B1), which in turn was found equal to the coefficient of $A = 1.5$ in Fairall et al. (2011). These results agree with Lorke and Peeters (2006) who derived a unified relation for interfacial fluxes (air-water and water-sediment) through a linear relationship of $u_{,wat}$ to $k$,

$\quad$ especially at the bottom interface where shear is the only relevant process. Furthermore, Equation 6 has a similar (i.e., quasi-linear) wind-$k$ relationship as the data-driven parameterization from *VP13* but cannot explain the higher order polynomial relationship reported in *CW03* and *CC98*.

**2.4.2 Convection**

A second source of dissipation at the surface is the convection ($\varepsilon_c$) resulting from surface cooling. The combination of wind

$\quad$ shear and free convection near a boundary is described by the Monin-Obukhov similarity theory (MOST) with a general form derived from a turbulent kinetic energy balance (Lombardo and Gregg, 1989; Tedford et al. 2014):

$$\varepsilon(z) = \varepsilon_u(z) \left( c_u + c_c \lfloor \frac{z}{L_{MO}} \rfloor \right), \qquad (8)$$

where $L_{MO}$ is the Monin Obukov length scale defined as $L_{MO} = u_{,wat}^3 / \kappa B_0$, including $\varepsilon(z) = c_u \cdot \varepsilon_u + c_c \cdot \varepsilon_c$ in Eq. (2) that can be rearranged as:

$\quad k_{NB} = a_1 \left( \varepsilon_u (c_u + c_c \cdot B_0/\varepsilon_u)^{1/4} \right) Sc^{-1/2}$ , $\qquad$ (9)

$a_1$ ranges in the literature from 0.2 to 1.2 (Soloviev et al., 2007; MacIntyre et al., 2010; Tedford et al., 2014; Heiskanen et al. 2014; Winslow et al., 2016), $c_u$ from 0.84 to 1 (Winslow et al., 2016) and $c_c$ from 0.37 to 2.5 (Wyngaard and Coté, 1971; Tedford et al., 2014). Hereafter, we use the following set of values: $c_u = 1$ and $c_c = 1$. Fairall et al. (2011) used an essentially equivalent approach but formulated in terms of a Richardson number to describe the partitioning between dissipation from

$\quad$ convection and wind shear, expressing the wind shear in terms of the air-side friction velocity: $R_f = B_0 v / u_{,atm}^4$, which can be integrated into (9) as:

$$k_{NB} = a_1 \left( \frac{\rho_{atm}}{\rho_{wat}} \right)^{1/2} u_{,atm} \left( \frac{c_u}{\kappa c} \left( 1 + \frac{R_f}{R_{f,c}} \right) \right)^{1/4} \left( \frac{Sc}{600} \right)^{-1/2} , \qquad (10)$$

with $R_{f,c} = \frac{c_u \rho_{atm}^2}{c_c \rho_{wat}^2 \kappa c}$. The details of this demonstration can be seen in Soloviev and Schlüssel (1994).



### 2.4.3 Wave action

The effect of surface waves is commonly implemented in oceanography but barely considered in limnology. All process-based models rely on the same parameterization of energy dissipation by wind shear and convection. They however differ in how they parameterize energy dissipation by wave action and wave breaking.

The contribution of the wave action ($\varepsilon_w$) is, accordingly, added as a third source of turbulence (Fig. 1). In the presence of
surface waves, the balance between $\tau_t$ and $\tau_0$ does not hold anymore. Therefore, Soloviev and Schlüssel (1994) added a corrective factor, $\varphi$, using the Keulegan number ($Ke = u_{,wat}^3/(gv)$), in order to decrease the component $\tau_t = \tau_0 \cdot \varphi$ where $\varphi$ $= 1/(1 + Ke/Ke_c)$, with the critical Keulegan number ($Ke_c$) define in Soloviev and Lukas (2006). As a result, the equation for shear-driven dissipation $\varepsilon_u(z)$ is:

$$\varepsilon_u(z) = \frac{u_{,wat}^4}{\kappa cv} \cdot \varphi^2 \, , \tag{11}$$

Following this step, the turbulent kinetic energy dissipation rate from wave action ($\varepsilon_w$) is added and defined with the Keulegan number by Soloviev et al. (2007) as:

$$\varepsilon_w = \alpha_W \left(\frac{3}{BS_q}\right)^{1/2} \frac{(Ke/Ke_{cr})^{3/2}}{(1+Ke/Ke_{cr})^{3/2}} \frac{u_{,wat}g}{0.062\kappa C_T(2\pi A_w)^{3/2}} \frac{\rho_{atm}}{\rho_{wat}} \, , \tag{12}$$

where $C_T = (z_0/H_s)$. $z_0$ is the surface roughness scale from the water side and the $C_T$ value is set as a constant at 0.6 (More details in Soloviev et al., 2007). This definition does not hold for closed basins because, in the case of incompletely developed
waves, the dissipation of energy from wind shear transmitted to the waves is not fully redistributed in the water body (Simon, 1997). Hence, for the application in Lake Geneva, we followed Terray et al. (1996) who defined a varying $C_T$:

$$C_T = 1.38 \cdot 10^{-4} \left(\frac{U_{10}}{C_p}\right)^{2.66} \, , \tag{13}$$

where $C_p$ is the peak speed of the wave spectrum defined in Deike and Melville (2018) according to Toba (1972, 1978). This leads to $C_T << 1$. This allows to increase the effect of $\varepsilon_w$ (inversely proportional to $C_T$ in Eq. 12) on $k$. Here, we used this
formulation to adapt the *S07* ocean model for a large lake (closed basin). Henceforth we refer to the adapted uncalibrated and calibrated models as *SD20* and *SD20-fit*, respectively (Table 1). Finally, these three terms of $\varepsilon$ ($\varepsilon_u$, $\varepsilon_c$, $\wedge \varepsilon_w$) can be added before computing the SRM (Eq. 2) for determining $k$-no bubble ($k_{NB}$).

### 2.4.4 Bubble enhancement

Additional deviations from the linear relationship to $U_{10}$ are explained by the gas transfer resulting from bubbles and sprays
during wave breaking. This mechanism is accounted for by adding a $k$-bubble ($k_B$) term to the already mentioned $k_{NB}$. Soloviev et al. (2007) used the empirical $k$-bubble parameterization from Woolf et al. (1997):





$$k_B = W \frac{2450}{O_s\left(1+\frac{1}{(14O_s Sc^{-0.5})^{1/1.2}}\right)^{1.2}} , \tag{14}$$

where $W$ is the fractional whitecap coverage only expressed as a function of wind ($3.84 \cdot 10^{-6} \cdot U_{10}^{3.41}$ and $O_s$ is Ostwald gas solubility. This formulation does not take wave height into account (Fig. B1). Nevertheless, a recent study (Deike and Melville, 245 2018) performed a new numerical process-based parameterization for gas transfer velocity from bubble enhancement considering $H_s$ through the following equation:

$$k_B = \frac{A_B}{O_s} u_{*,atm}^{5/3} (gH_s)^{2/3} \left(\frac{Sc}{600}\right)^{-1/2} , \tag{15}$$

where $A_B$ is an empirical factor with dimension (= $10^{-5}$ m$^{-2}$ s$^2$) and $O_s$ defines by the ideal gas constant ($R$), the surface water temperature ($T_0$) and, $CO_2$ solubility coefficient in freshwater ($\alpha$) (Reichl and Deike, 2020). Then, the gas transfer velocity is 250 expressed as a sum of no-bubble $k_{NB}$ and bubble $k_B$ components (Table 1: *S07, DM18, SD20, and SD20-fit*) following Keeling (1993) and Woolf et al. (1997, 2005). Our adapted models modified from *S07* (*SD20* and *SD20-fit*) include a refined parameterization of the wave action term $\varepsilon_w$ along with the bubble term from *DM18* (*SD20* and *SD20-fit*). In addition, for the model *SD20-fit*, the $a_1$ parameter of Eq. (2) and the $A_B$ parameter of Eq. (15) were fitted to the $k_{600}$ observations ($a_1 = 0.33$ and $A_B = 3\ 10^{-5}$ m$^{-2}$ s$^2$).


With this review of existing and adapting parameterizations, we show that (i) there is a discrepancy between SRM-based model with shear stress as the only energy source and empirical parameterizations with polynomial (order > 1) wind-based relationship. Such a discrepancy is tentatively resolved by adding the effect of convection and surface waves. (ii) We further highlight that most fitting parameters from the different SRM-based models are in good agreement. (iii) We finally recall that 260 it is possible to provide a unifying parameterization of $k$ with SRM model including wind shear, wind-induced waves, and convection with only a few input parameters such as $U_{10}$, $B_0$, and *Fetch*.

## 3 Results

### 3.1 Observed and predicted $k$

After quality check, our dataset contains 94 discrete $CO_2$ flux observations. We first assess the representativeness of our 265 sampling by comparing the survey-specific and annual distributions of the three main inputs for $k$-models: $U_{10}$ (all models), $B_0$ during convective periods (*T14, S07, SD20 & SD20-fit*) and $H_s$ (*S07, DM18, SD20 & SD20-fit*) (Fig. 3; Temporal evolution of these three terms in Fig. C1). From 13$^{th}$ June 2019 to 12$^{th}$ June 2020, the average wind speed over Lake Geneva is 2.9 m s$^{-1}$ with a mode at 2.5 m s$^{-1}$; very low wind speeds (< 1 m s$^{-1}$) are encountered 12 % of the year, while high- (> 5 m s$^{-1}$ to very high > 10 m s$^{-1}$) wind events represent 15 % and 2 % of the year, respectively. The sampling surveys covered the full annual 270 range of $U_{10}$. Average and modal values of $B_0$ over the year are close to 0.25 10$^{-7}$ m$^2$ s$^{-3}$. However, the sampling covered only





the lowest 50 % of the annual distribution and under-samples conditions of potentially strong convection. Considering that the dissipation by buoyancy flux, as parameterized in the process-based models, is already well known in the literature and that it is not the central point of our study, we posit that the under-sampling of $B_0$ is therefore not expected to significantly affect our analysis. The predicted modal $H_s$ value is 0.15 m over the year. Events of high $H_s$ (> 0.4 m) represent 6 % of the year, with a

maximum $H_s$ of 1.1 m. As for $U_{10}$, the surveys covered the full range of annual $H_s$.

**Figure 3**

Observationally based $k_{600}$ are shown with their error bars corresponding to the uncertainties of the pCO$_2$ in air and in water (± 50 ppm) in Fig. 4a. We notice that all the measurements with a wave height > 0.4 m were observed for wind speeds > 5 m s$^{-1}$ and the corresponding $k_{600}$ are located above the linear function (i.e., from a linear regression against wind shear velocity;

Fig. B1) scaling $k_{600}$ to $u_*$ (i.e., first order relationship) We then compare the $k_{600}$ observed during the specific surveys to the values computed with all $k_{600}$ models throughout the annual cycle, in relation to $U_{10}$ (Fig. 4b-c). Table 2 provides the root-mean square errors (RMSE) for all model estimates compared to $k_{obs}$ during the flux surveys, (i) for the full dataset (All Wind), and split (ii) for low wind (< 5 m s$^{-1}$, LW) and (iii) strong wind conditions (≥ 5 m s$^{-1}$, SW). The three empirical wind-based models only depend on wind (Fig. 4b). Both *CC98* and *CW03* were originally calibrated for small lakes, using a mass balance

calibration method (Table 1). However, they lead to divergent gas transfer velocities, particularly above 5 m s$^{-1}$, illustrated by a RMSE for SW as high as 22.8 cm h$^{-1}$ for *CC98* while *CW03* performs better (RMSE SW = 12.8 cm h$^{-1}$). Furthermore, both models underestimate $k_{600}$ at low wind (Fig. 4a), with a higher deviation for *CW03* (Table 2). The $k$ values predicted by *VP13* are closer to those of the process-based models that explicitly integrate wave actions (*S07* and *DM18*) (Fig. 4bc), demonstrating that lake size integration in the empirical model captures at least part of the wave action on $k$. Performances of *VP13* at strong

winds (RMSE SW = 12.7 cm h$^{-1}$) were better than those of the ocean-derived models integrating surface waves (RMSE SW = 13–15.9 cm h$^{-1}$). However, *VP13* shows a positive offset during calm periods, along with the highest RMSE of the set of models at low wind speed (Fig. 4b; Table 2).

**Figure 4**

The process-based models (Fig. 4c) provide different $k_{600}$ values for a given wind speed, owing to the integration of additional

environmental components (i.e., the varying drag coefficient, the convective mixing in *R12* and *T14* as well as the effect of waves in *S07*, *DM18*, *SD20* and *SD20-fit*). All process-based models are similar at low winds as they share a common physical basis for parameterization of wind shear and convection. Therefore, they lead to similar RMSE (2.9–3.5 cm h$^{-1}$) under such conditions, where surface waves are negligible (Table 2). Divergences occur at higher wind speeds. *T14*, initially developed for small lakes with limited wind exposure, performed the worst (RMSE: 19.8 cm h$^{-1}$). This increased $k$-underestimation at

high winds can be attributed to (i) dissipation by wave action and bubble formation not considered (in *R12* and *T14*), and (ii) to the use of a constant $z(0) = 0.15$ m in the *T14* model (Eq. 5). This approximation of the diffusive layer is consistent with low wind speed but is almost one order of magnitude too large under strong wind speed. Other process-based models, designed for greater wind range (> 10 ms$^{-1}$), integrate surface waves and, as a result, lead to better estimates than *R12* and *T14* (RMSE: 10.4–15.9 cm h$^{-1}$). However, the ocean wave model of *DM18* shows lower performances at strong winds than *CW03* and *VP13*





(Fig. 4bc). Finally, the specific fit parameterization of the *SD20-fit* model improves the performance at high wind speeds by
~30 % (RMSE = 10.5 cm h$^{-1}$), outperforming all the other methods.

**Table 2**

### 3.2 Surface wave integration

We herein scrutinize how those varying parameterizations ultimately alter the shape of relationship between $k_{600}$ and $U_{10}$. In
*R12* (Fig. 5a), wind is only included through wind shear, resulting in a linear relationship between $k_{600}$ and $U_{10}$, as already
anticipated. Adding the wave action (no bubble) through the *S07* parameterization (Fig. 5b) does not lead to any significant
departure from the minimal *R12* model. Adapting wave action by decreasing $C_T$ (for the no-bubble term) leads to a departure
from the wind shear linear relationship for $H_s > 0.4$ m (Fig. 5c).

Then, adding the $k$ bubble term related to wave breaking of *DM18* further increases this deviation from the linear $k_{600}$-$U_{10}$
relationship but also scatters $k_{600}$ estimates for a given $U_{10}$ (Fig. 5d). Finally, the fitting with observationally based $k_{600}$ improves
the estimation for strong wind (Fig. 5e; Table 2). Given the range of wind fetch from ~ 0.5 km to ~30 km, the contribution of
waves varies for a given wind speed depending on the fetch, as evidenced by the scattering of the parameterized $k_{600}$ for a
given $U_{10}$ (Fig. 5f). A significant modification of $k_{600}$ by wave action and wave breaking occurs for a fetch length > 15 km and
$U_{10} > 5$ m s$^{-1}$ (Fig. 5f), in the case of Lake Geneva, generating wave of $H_s > 0.4$ m (Fig. 5e).

**Figure 5**

As compared to $k_{600}$ estimated by *R12* (Fig. 5a), the *SD20* and *SD20-fit* models provide $k$ estimates that are 20–50 % higher
for $U_{10} = 10$ m s$^{-1}$, respectively, and 40–70 % higher for $U_{10} = 15$ m s$^{-1}$. Therefore, adapting the surface waves, through the
change of the wave action for incompletely developed waves and the fitting to observed data encountered in local lake
conditions, leads to better performances of *SD20* models. *SD20-fit* reached the lowest RMSE at all wind speeds and is thereafter
used as a reference for the modelling of the annual gas transfer velocities.

### 3.3 Annual cumulative gas transfer velocity and the effect of extreme conditions

We show above that models accounting for surface waves better represent the non-linear increase in $k_{600}$ at high winds. Because
high-wind events remain rare, we test whether a better representation of $k_{600}$ during rare, high-wind events affects the local
estimates of $k_{600}$ over a full year. To this end, cumulative sums of hourly $k_{600}$ were computed over a full annual cycle (13$^{th}$ June
2019 – 12$^{th}$ June 2020) for all $k$-models (Fig. 6). The annual dynamics such as the annually averaged $k_{600}$ were compared using
*SD20-fit* as a new reference model.

**Figure 6**

Cumulated $k_{600}$ computed for *SD20-fit* shows some episodic steep increases between December and March, due to the winterly
prevalence of high wind events (winter average wind speed, 3.25 m s$^{-1}$, was greater than the summer mean, 2.55 m s$^{-1}$, by 25
%) and greater significant wave height (winter average wave height, 0.15 m, greater than the summer value, 0.10 m, by 50 %)





(Fig. C1). The average hourly $k_{600}$ by the *SD20-fit* model is 7.3 ± 7.4 cm h$^{-1}$ (mean ± se, Fig. 6a). Periods of high-winds, although accounting for 15 % of data points, contribute 44 % of annually cumulated-$k_{600}$ in the *SD20* and *SD20-fit* models (Fig. 6b), while the periods of high waves ($H_s \geq 0.4$ m) accounting for only 6 % of data points, contribute to more than 20 %

of annually cumulated-$k_{600}$. The wind-based models are those for which cumulative $k_{600}$ diverges the most from the *SD20-fit* reference model, with the lowest annual averaged $k_{600}$ for *CC98* and *CW03* (3.9 ± 2.7 and 4.8 ± 9.3 respectively) and the highest for *VP13* (9.9 ± 6.1). These divergences arise from the low performances of these models at low wind regimes (Fig. 4a, 6b; Table 2), which represent 85 % of annual data-points. All the other process-based models have relative dynamics of cumulative $k_{600}$ similar to that of the *SD20-fit* model and end up with annually averaged $k_{600}$ that are 15 % lower than for the

*SD20-fit*. The representation of $k_{600}$ at low wind speeds is similar for all process-based models, and the divergence arises from the representation of the rarer high wind speed episodes, which contribute to 43–46 % of annual cumulative $k_{600}$ (Fig. 6b).

## 4 Discussion

The history of *k*-models, simulating the gas transfer velocity for surface waters, dates back from the early 1990's. *k*-models have been developed and tested in small lakes sheltered from winds (e.g., Crusius and Wanninkhof, 2003; Tedford et al.,

2014), large lakes under low to moderate wind speed (Vachon and Prairie, 2013), and oceans (e.g., Soloviev et al., 2007; Fairall et al., 2011; Esters et al., 2017; Deike and Melville, 2018). While the effects of surface waves on *k* can be neglected in small lakes, we question herein whether this assumption holds for large lakes such as Lake Geneva, in which surface waves are frequently observed (Fig. 2; Fig. C1). We evaluated the performance of different experimental-based and process-based models to estimate $k_{600}$ in the large Lake Geneva. We show that integrating the effect of wave formation at high wind speeds and long

fetch better represents the sharp increase of the $k_{600}$ values during such episodic windy events.

### 4.1 Choice of *k*-models

Wind-based models have been long known for misestimating $k_{600}$ at low wind speeds (Eugster et al., 2003; MacIntyre et al., 2010; Erkkilä et al., 2018). Consistently, wind-based models showed the lowest performances for Lake Geneva, especially at low wind speeds (*CW03* and *VP13*), which resulted in large discrepancies in annually averaged and cumulative $k_{600}$ over the

full year. They are however easy to compute, require few inputs (only $U_{10}$) and, remain by far the most used to estimate lakes $CO_2$ emissions worldwide (e.g., Raymond et al. 2013). One solution to increase the performance of wind-based models is to revise calibration at each new site (Klaus and Vachon, 2020). Another possibility is to broadly adopt process-based models. The presented process-based models require input data that are today more easily accessible: wind speed, heat flux and wind fetch (i.e., distances from the shore) routinely acquired at high-frequency in many lakes. The development of R packages such

as Lake Metabolizer (Winslow et al., 2016) in which the calculations of process-based models are implemented, also alleviates their computational difficulty. Both increased data availability and computational tools should foster the use of process-based *k*-models, which hold great potential to obtain more accurate global $k_{600}$ estimates.





The analysis of the models adapted from the existing literature to account for the effect of waves, *SD20* and *SD20-fit* (Fig. 5d-

e-f), showed that the wave contribution to $k$ becomes significant for $H_s > 0.4$ m, corresponding, for Lake Geneva, to winds blowing at 5 m s$^{-1}$ from the southwest where the fetch length is maximal (> 15 km) with respect to the measurement site. A significant to the gas transfer velocity by surface waves is expected in lakes where $H_s > 0.4$ m is not infrequent. Wave heights beyond this threshold value of $H_s$ are frequently encountered in lakes of similar or greater sizes than Lake Geneva (6 % of annual time in Lake Geneva). In the Great Lakes of North America, Hubertz et al. (1991) showed that the mean wave height

of all these lakes were > 0.4 m in summer and close to 1 m in winter with a maximum of up to 5 m. $H_s > 0.4$ m can also form over elongated lakes of smaller size, such as smaller Swiss Lakes (e.g., Lake Neuchâtel, Lake Bienne) (Amini et al., 2017). Since *SD20-fit* is a process-based model integrating the four main processes in a mathematically coherent way, we would expect that it can be applied to such lakes experiencing $H_s > 0.4$ m and improves the accuracy of $k$ estimates. Because waves can physically damage in-shore and off-shore infra-structures, many large lakes benefit from wave forecasts. $H_s$-data from

those forecasting systems (e.g., National Data Buoy Centre – NOAA, Wave Atlas from SwissLakes.net; Amini et al., 2017) could allow testing whether the *SD20-fit* models can be applied to those lakes and whether $k_{NB}$ and $k_B$ through $a_1$ and $A_b$ need to be recalibrated or fitted to the local context if flux measurement data are available, as for this study. Energy dissipation during high-wave events increases the gas-transfer velocity well beyond the linear relationship derived for wind shear alone. We therefore expect that computed gas fluxes at the air-water interface should be significantly improved by the integration of

surface waves into the $k$-models.

## 4.2 Implication of four components on the annual $k$ estimation and the annual $CO_2$ fluxes

### 4.2.1 Seasonal distribution of $k_{CO2}$

Converting $k_{600}$ to $k_{CO2}$ using the Schmidt number (Wanninkhof, 1992) highlights the importance of water temperature in gas exchange dynamics. Indeed, the seasonal distribution of the cumulative $k_{600}$ is ~20 % and ~30 % for the warm (spring and

summer) and cold (autumn and winter) seasons, respectively. Once the temperature-effect accounted for, this distribution increases to 26.1 % for the summer and decreases to 24.9 % for the winter but remains unchanged for spring and autumn. While *R12* only use wind shear and convective terms, the selected process-based model (*SD20-fit*) allows a decomposition into the four main drivers of the gas transfer velocity, hence paving the way to a better understanding of the implication of these processes throughout an annual cycle.

**Figure 7**

Wind shear remains the dominant component of the gas exchange velocity over the different seasons (Fig. 7a). The annual contribution of surface waves (wave action and bubble formation) is limited to 9 to 10 % of the cumulative $k$ in Autumn and Winter. The contribution of buoyancy flux at surface to $k$ is even smaller for both models (*R12* and *SD20-fit*) at this seasonal scale. Yet, both the buoyancy flux and the surface waves can significantly increase $k$ during episodic events, during which





they can contribute disproportionately to $k$ at hourly (up to 80% for convection) and daily (up to 25 % for surface waves) time

scales (Fig. 7b). Several studies have emphasized the disproportionate contribution of episodic mixing events on annual flux,

bringing $CO_2$ back to lake surfaces such as after ice break in dimictic lakes (Karlsson et al., 2013; Finlay et al., 2019) or during

fall mixing on a eutrophic deep lake (Reed et al., 2018). Process-based $k$-models integrating both the buoyancy flux and the

wind-induced waves offer the opportunity to mechanistically investigate how much those episodic events contribute to annual

emissions through short-term modifications of the gas exchange velocity.

**4.2.2 Consequences on the choice of $k$-model on the monthly to annual $CO_2$ flux estimation**

Monthly fluxes were computed based on $k$-estimates from the different models at hourly timestep and the monthly average of

water temperature and recorded $pCO_2$ at the lake surface (OLA-IS, AnaEE-France, INRAE of Thonon-les-Bains, CIPEL,

Rimet et al., 2020; Perga et al., 2016) as well as a constant $pCO_2$ in the atmosphere (400 µatm). As predicted by the Fick's

law, the highest outgassing fluxes occur in fall and winter, when water mixing brings $CO_2$ up to the lake surface, while low

up-taking gas fluxes occur in spring and summer, when primary production depletes surface $CO_2$ below saturation. However,

annual estimates of net $CO_2$ outgassing vary from 14.7 to 37.1 mmolC m$^{-2}$ d$^{-1}$ (Table 3) depending on the $k$-model used for

computation. Consistently, differences between model estimates are relatively low in summer since both the $\Delta pCO_2$ gradient

(100-200 µatm) and wave occurrence are limited. Estimated fluxes are strongly dependent on the chosen $k$-model in winter

when both $\Delta pCO_2$ (475 µatm) and surface waves occurrence are higher (Table 3; Fig.7). Therefore, while high wave events

represent only 6 % of the total surface waves occurrence ($H_s > 0.4$ m), an incomplete consideration and description of their

contribution may lead to an annual flux underestimation of about 20–25 %. The weak contribution of convection is at odds

with observations in small lakes, but not unexpected, since large lakes are exposed to stronger winds, such that wind shear-

driven $\varepsilon_u$ often outpaces convectively driven, $\varepsilon_c$ (Read et al, 2012). However, the limited impact of the buoyancy flux on $k$

does not rule out its contribution to $CO_2$ exchange. Indeed, convective mixing plays a central role in the deepening of the

mixed layer allowing the export of the $CO_2$ stored in the hypolimnion towards the surface during the cold period and thereby

controlling the $pCO_2$ gradient (Zimmerman et al, 2020) and the observed wintertime outgassing. Altogether, both surface

oversaturated $CO_2$ concentrations (as a result of convective mixing) and wind-induced waves are more relevant in fall and

wintertime for the monomictic Lake Geneva, leading to most of the annual outgassing during this season (Table 3). As for

many monomictic lakes, these seasons drive most of the annual $CO_2$ budget of Lake Geneva (Perga et al, 2016), while they

usually correspond to those where direct measurements are the scarcest. An improved quantification of $k$-values through SRM-

models including wind-induced waves should contribute to refining the overall estimation of large lakes contribution to

regional $CO_2$ emissions.

**Table 3**

## 5 Conclusion

Investigations of the four main processes generating the gas transfer velocity in the large Lake Geneva demonstrated the importance of considering surface waves during episodic windy events responsible for more than 44 % of annual cumulated $k_{600}$. The in-depth study of the behaviour of the process-based models has enabled to underscore their consistent predictions at low and strong wind, especially considering the new combination and adaptation model, *SD20-fit*. This last model significantly improves the estimation of $CO_2$ flux when these three thresholds appear in the field: $U_{10} > 5$ ms$^{-1}$, Fetch > 15 km, and $H_s > 0.4$ m, making it applicable in a wide range of lake sizes. Furthermore, *SD20-fit* is assembled on solid theoretical bases coming from limnological and oceanic literature and allows to analyse the distribution of these four main terms ($k_u$, $k_c$, $k_w$, and $k_b$) across a variety of time scales depending on the kind of study.

Noteworthily, *SD20* was built on the basis of a single measurement point on the lake, just as for most of the existing $k$ models. Therefore, the question of the extrapolation of the model to the whole of the lake remains essential. We assume three ways of different complexities: (i) estimate an average fetch value depending on the wind direction and the geometry of the lake, (ii) discretize the lake into a few parts according to the complexity of its geometry and direction of the prevailing winds, and (iii) discretize the lake into a large number of pixels based on 2D or 3D wind models available in some countries in order to estimate gas transfer velocity and gas fluxes at a finer spatial scale. Nevertheless, the question of the spatial variability of the $\Delta CO_2$ is still open and difficult to analyse at high frequency in large lakes.

To conclude, this study sheds light on the complexity of large lakes located at the interface between small, sheltered lakes and the open oceans, thus experiencing a combination of processes relevant for both small and large systems. The possibility of using process-based models in a fairly simple way with few inputs to improve the precision of the gas transfer velocity and therefore the gas flux should be supported in future research. In addition, this approach is very promising regarding long-term trends of $CO_2$ emissions from lakes, as well as a finer estimation of fluxes during more intense episodic events.

**Appendix A**

**Figure A1**

**Figure A2**

**Figure A3**

**Appendix B**

**Figure B1**





**Appendix C**

**Figure C1**

**Data availability.** Water temperature, buoyancy flux at surface, and meteorological data are available in the datalakes – Open research data publishing platform (https://www.datalakes-eawag.ch). Water $CO_2$ concentration and $CO_2$ flux measurements are available upon request from the first author.


**Author contribution.** PP, MEP, and DB designed the study. PP, BFC, NE, and TL collected field data and carried out data pre-processing. PP developed the model code and performed the simulation with contribution from DB. PP, MEP, and DB drafted the manuscript and all co-authors contributed to the final submitted manuscript.

**Competing interests.** The authors declare that they have no conflict of interest.

**Acknowledgements.** We would like to thank the entire team from LéXPLORE platform, for their administrative and technical support and for LéXPLORE core dataset. We also acknowledge LéXPLORE five partner institutions: Eawag, EPFL, University of Geneva, University of Lausanne and CARRTEL (INRAE-USMB). This study was supported by 475 CARBOGEN project (SNF 200021_175530) linked to LéXPLORE project (SNF R'Equip, P157779). The authors thanks Sébastien Lavanchy, chief technical officer (APHYS-EPFL) and Aurélien Ballu, member of the technical pool (IDYST-UNIL) of LéXPLORE platform for their technical and field supports. B. F. C. was supported by the European Union's Horizon 2020 research and innovation program under the Marie Skłodowska-Curie grant agreement No. 834330 (SO-CUP).

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





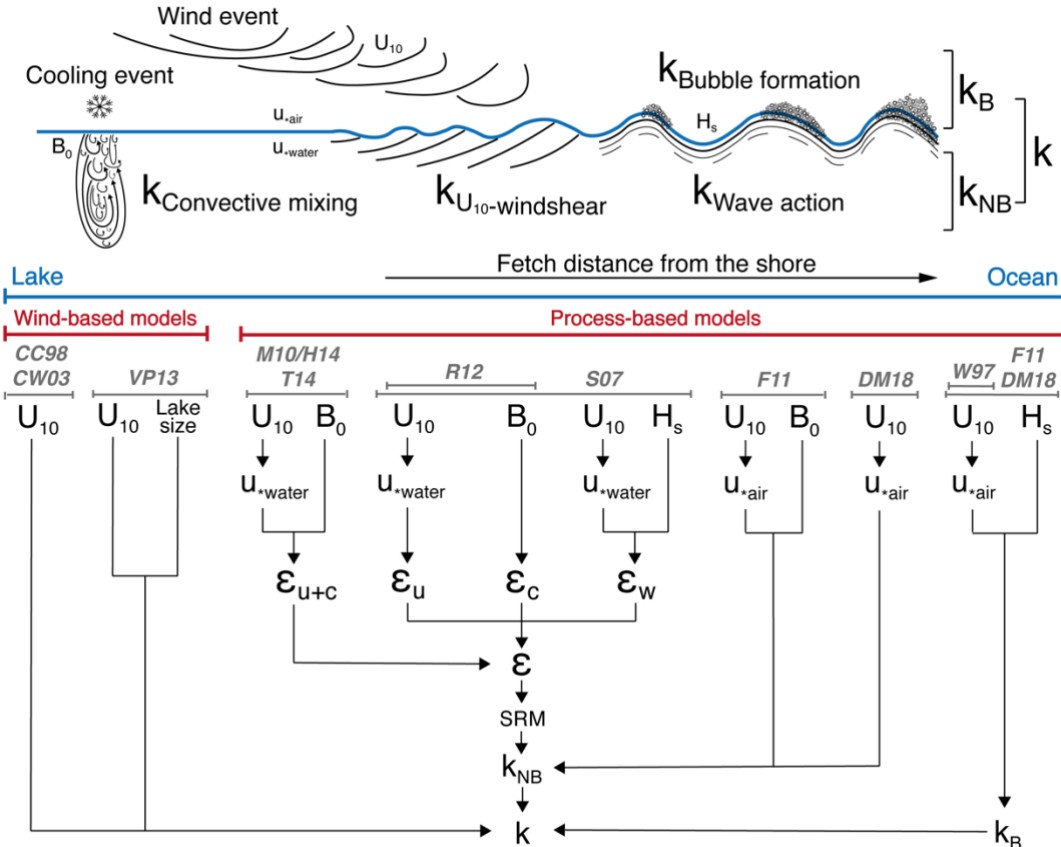

**Figure 1: Conceptual scheme of the four main processes driving gas transfer velocity ($k$) in a large lake induced by wind and cooling events. These four processes are split into two types of $k$: $k$-bubble for the bubble formation ($k_B = k_b$) and $k$-no bubble for the convective mixing, wind shear and wave action term which are added ($k_{NB} = k_c + k_u + k_w$). Below this scheme, a non-exhaustive review about conceptual approaches of k-models used in 1st Fickian law. From left to right, increase in the complexity level of $k$-models as well as their study site (limnological to oceanic case). All these variables are described in the section 2.4 and in Table 1.**




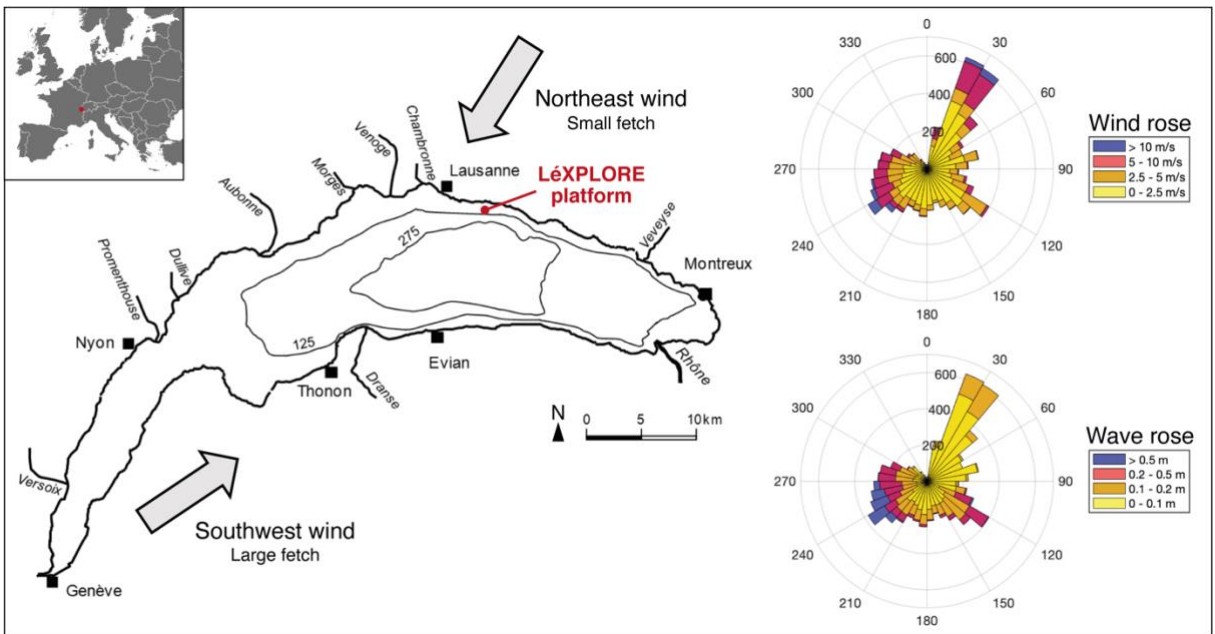

**Figure 2: Location and map of Lake Geneva with the two prevailing winds (left side) also depicted by the wind rose (top right side). The wave rose highlights the highest wavefield generated at the sampling location by the southwest wind with a larger fetch (bottom right side). Both wind and wave roses are computed with annual data from 13th June 2019 to 12th June 2020 at LéXPLORE.**

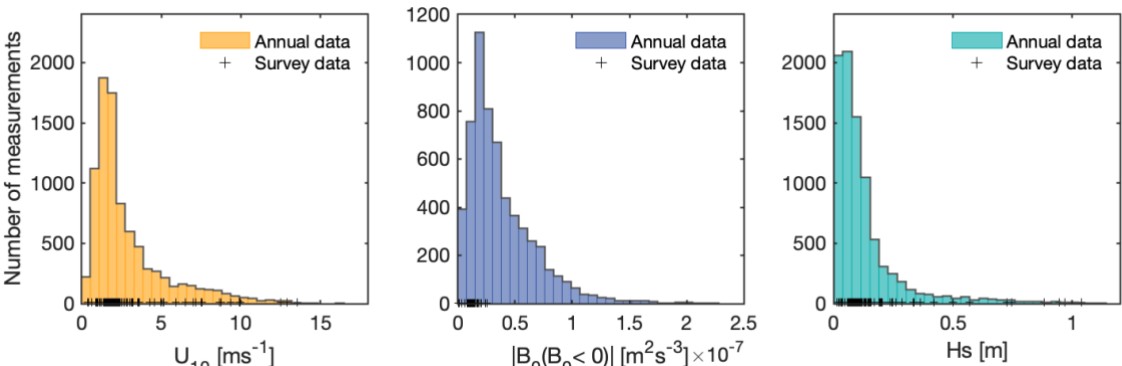

**Figure 3: Annual distribution of three main components used to compute $k_{600}$ models. Orange) Wind speed at 10 m; Blue) Buoyancy flux at surface during cooling; Turquoise) Significant wave height; and these survey data observed during $CO_2$ flux measurements after quality control (+).**





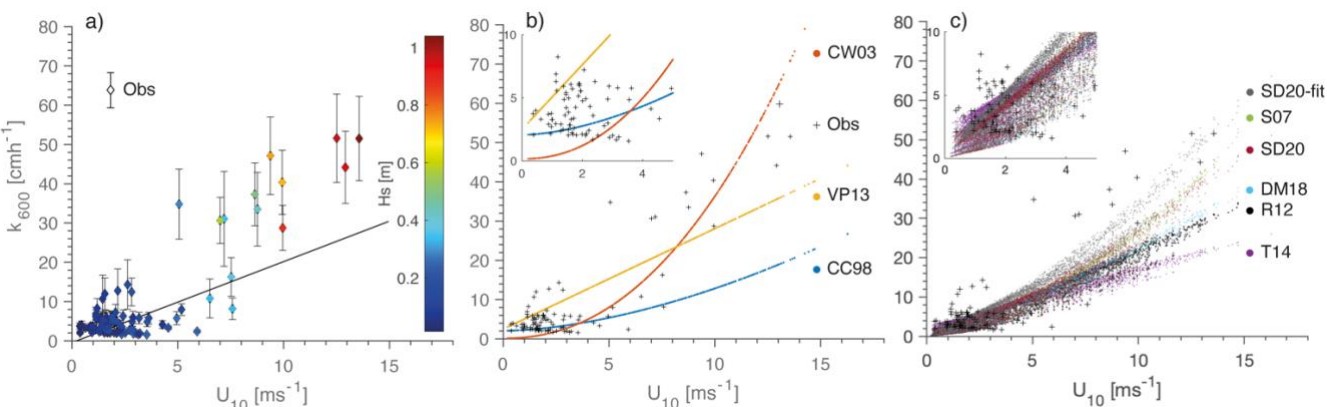

**Figure 4: a)** $k_{600}$ **observed as a function of** $U_{10}$ **and coloured according to** $H_s$ **(colorbar), and the error bars produced by the uncertainty of** $\Delta CO_2$ **(± 50 ppm) as well as the** $u_*$-$k_{600}$ **linear regression (solid line: see also Fig. B1); b)** $k_{600}$ **wind-based models (***CC98***,** ***CW03*** **&** ***VP13***); c)** $k_{600}$ **process-based models (***T14***,** ***S07***,** ***DM18***,** ***SD20*** **&** ***SD20-fit***) computed with annual data; Observed** $k_{600}$ **derived from** $CO_2$ **flux chamber measurements (+).**

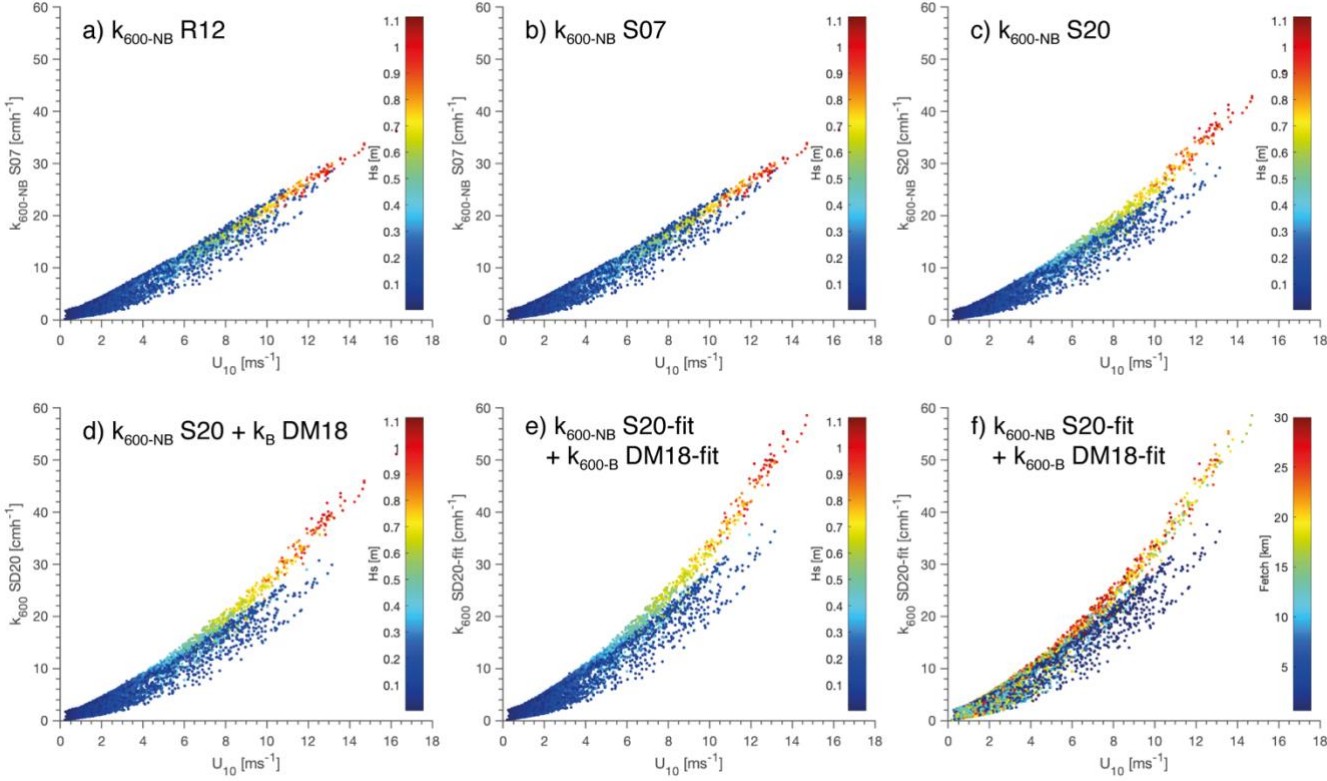

**Figure 5: Relation** $U_{10}$ **vs** $k_{600}$ **modelled and coloured according to** $H_s$ **(colorbar) in a-b-c-d-e as well as coloured according to fetch distance (colorbar) in f. a)** ***R12*** **integrating wind shear and convection; b)** ***S07*** **integrating wind shear, convection, and wave action for fully developed waves; c)** ***S20*** **integrating wind shear, convection, and wave action for not fully developed waves; d)** ***SD20*** **similar to** ***S20*** **adding the** $k$ **bubble term of** ***DM18***; **e-f)** ***SD20-fit*** **similar to** ***SD20*** **with** $a_1$ **and** $A_b$ **fitted to** $k$ **observed.**






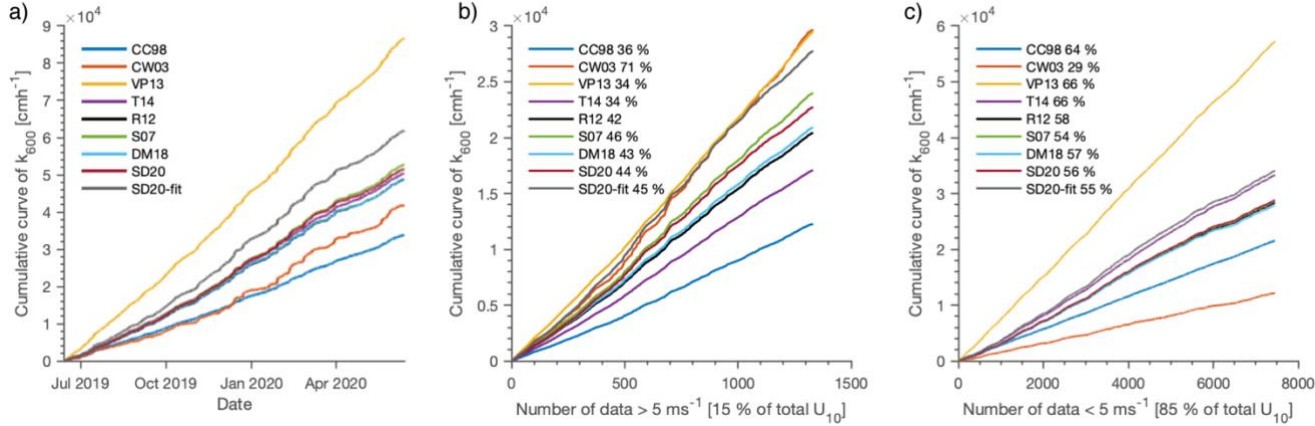

**Figure 6: a) Cumulative $k_{600}$ modelled over an annual cycle; b) Cumulative $k_{600}$ for wind $< 5$ m s$^{-1}$; c) Cumulative $k_{600}$ for wind $\geq 5$ m s$^{-1}$.**

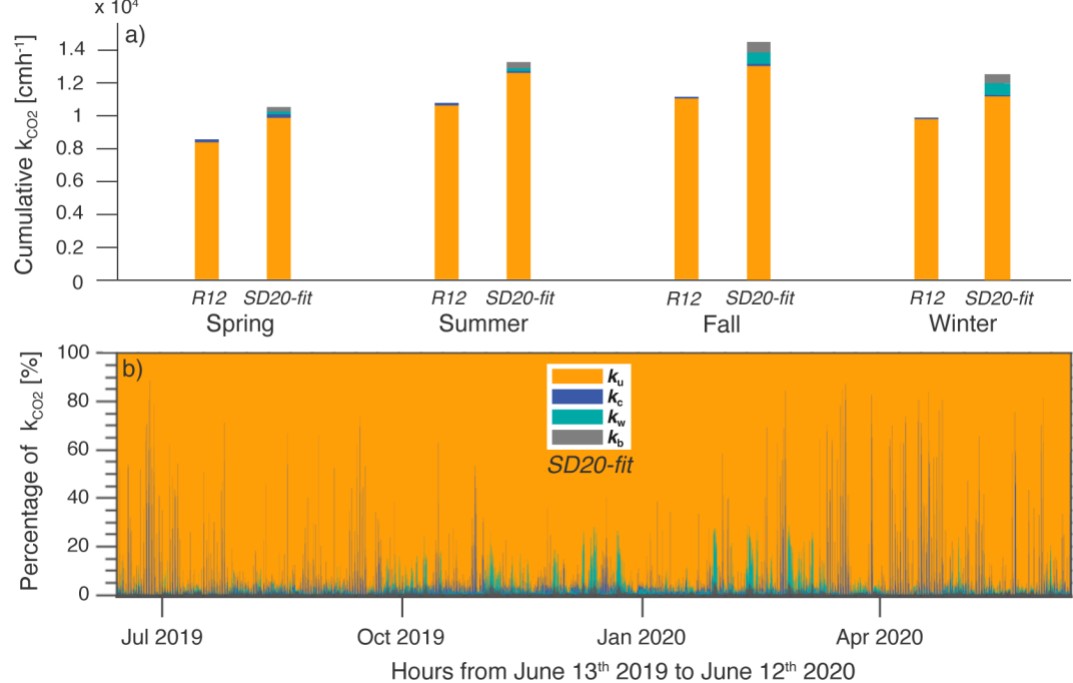

**Figure 7: a) Distribution of $k_{CO2}$ generated by two main processes ($k_u$ and $k_c$) in *R12* and four main processes ($k_u$, $k_c$, $k_w$, and $k_b$) in *SD20-fit* for each season; Spring (April-May-June); Summer (July-August-September); Fall (October-November-December); Winter (January-February-March). The height of bar represents the cumulative of $k_{CO2}$ by season for both models (*R12* and *SD20-fit*); b) Distribution of four $k$ generated by wind shear, convection, wave action and bubble enhancement ($k_u$, $k_c$, $k_w$, and $k_b$) along the annual cycle. Use of *SD20-fit* model where $k_u = SRM(\varepsilon_u)$, $k_c =$**
**$SRM(\varepsilon_u + \varepsilon_c + \varepsilon_w) - SRM(\varepsilon_u + \varepsilon_w)$, $k_w = SRM(\varepsilon_u + \varepsilon_c + \varepsilon_w) - SRM(\varepsilon_u + \varepsilon_c)$, and $k_b$.**



**Table 1: Summary of characteristics of $k_{Sc}$ models for predicting the air-water gas transfer velocity based on wind speed (*CC98, CW03*) and lake size (*VP13*), surface renewal model (*T14, R12 & S07*), COARSE approach (*DM18*) and both adapted models called *SD20* and *SD20-fit* from a combination of *S07* and *DM18*.**

| Model | Equation | Method | Site | Calibrated range |
|---|---|---|---|---|
| CC98 | $k_{600} = 2.07 + 0.215 \cdot U_{10}^{1.7}$ <br> $k_{Sc} = k_{600}\left(\dfrac{Sc}{600}\right)^{-1/2}$ | Mass Balance By gas tracer | Lake | Area (0.15 km²) <br> $U_{10}$ (< 9 ms⁻¹) |
| CW03 | $k_{600} = 0.168 + 0.228 \cdot U_{10}^{2.2}$ <br> $k_{Sc} = k_{600}\left(\dfrac{Sc}{600}\right)^{-1/2}$ | Mass Balance By gas tracer | Lake | Area (0.128 km²) <br> $U_{10}$ (< 6 ms⁻¹) |
| VP13 | $k_{600} = 2.51 + (1.48 \cdot U_{10}) + (0.39 \cdot U_{10}$ <br> $\cdot \log 10(Lake\ size)$ <br> $k_{Sc} = k_{600}\left(\dfrac{Sc}{600}\right)^{-1/2}$ | Floating Chamber | Lakes | Area (0.2–602 km²) <br> $U_{10}$ (< 6 ms⁻¹) |
| T14 | $k_{Sc} = a_1 \cdot (\varepsilon \cdot v)^{1/4} \cdot Sc^{-1/2}$ <br> $\varepsilon = \varepsilon_{Wind\ shear + Convection}$ | Microstructure profiling | Lake | Area (4 km²) <br> $U_{10}$ (< 10 ms⁻¹) |
| R12 | $k_{Sc} = a_1 \cdot (\varepsilon \cdot v)^{1/4} \cdot Sc^{-1/2}$ <br> $\varepsilon = \varepsilon_{Wind\ shear} + \varepsilon_{Convection}$ | - | - | Following *S07* |
| S07 | $k_{Sc} = k_{Sc-NB-S07} + k_{Sc-B-W97}$ <br> $k_{Sc-NB} = a_1 \cdot (\varepsilon \cdot v)^{1/4} \cdot Sc^{-1/2}$ <br> $\varepsilon = \varepsilon_{Wind\ shear} + \varepsilon_{Convection}$ <br> $+ \varepsilon_{Wave\ action}$ | Eddy covariance | Ocean | Area (>100'000 km²) <br> $U_{10}$ (< 20 ms⁻¹) <br> Wave (0–10 m) |
| DM18 | $k_{Sc} = (k_{NB} + k_B) \cdot (Sc/600)^{-1/2}$ <br> $k_{NB} = A_{NB} \cdot u_{*,atm}$ <br> $k_B = (A_B / O_s) \cdot u_{*,atm}^{5/3} \cdot (g \cdot H_s)^{2/3}$ | Eddy Covariance | Ocean | Area (>100'000 km²) <br> $U_{10}$ (< 30 ms⁻¹) <br> Wave (1–10 m) |
| SD20 | $k_{Sc} = k_{Sc-NB-S07*} + k_{Sc-B-DM18}$ <br> *Adaptation of $\varepsilon_{Wave\ action}$ for large lake | Floating Chamber | Lake | Area (582 km²) <br> $U_{10}$ (< 16 ms⁻¹) <br> Wave (0–1.2 m) |
| SD20-fit | $k_{Sc} = k_{Sc-NB-S07*} + k_{Sc-B-DM18}$ <br> with $a_1$ from $k_{Sc-NB-S07*}$ and $A_B$ from $k_{Sc-B-DM18}$ fitted to observations | - | - | - |

*CC98* **Cole and Caraco (1998),** *CW03* **Crusius and Wanninkhof (2003),** *VP13* **Vachon and Prairie (2013),** *T14* **Tedford et al. (2014),** *R12* **Read et al. (2012),** *S07* **Soloviev et al. (2007),** *DM18* **Deike and Melville (2018).**





**Table 2: RMSE of $k_{600}$ models for all wind speed $(U_{10})$, $U_{10} < 5$ m s$^{-1}$ (i.e., LW) and $U_{10} \geq 5$ m s$^{-1}$ (i.e., SW).**

| RMSE | CC98 | CW03 | VP13 | T14 | R12 | S07 | DM18 | SD20 | SD20-fit |
|---|---|---|---|---|---|---|---|---|---|
| All $U_{10}$ | 9.8 | 6.5 | 6.7 | 8.6 | 7.5 | 6.2 | 7.3 | 6.2 | 5.2 |
| $U_{10} < 5$ ms$^{-1}$ | 3.2 | 4.2 | 4.5 | 2.9 | 3.3 | 3.3 | 3.5 | 3.3 | 3.2 |
| $U_{10} \geq 5$ ms$^{-1}$ | 22.8 | 12.8 | 12.7 | 19.8 | 16.6 | 13 | 15.9 | 13.1 | 10.5 |


**Table 3: Monthly to annual CO$_2$ flux estimation (mmolC m$^{-2}$ d$^{-1}$) from $k$-models and monthly $\Delta$CO$_2$ average (µatm) as well as their deviation from *SD20-fit*.**

| Period | $\Delta CO_2$ | CC98 | CW03 | VP13 | T14 | R12 | S07 | DM18 | SD20 | SD20-fit |
|---|---|---|---|---|---|---|---|---|---|---|
| April | 42 | 2.7 | 2.3 | 6.7 | 3.3 | 2.9 | 3.0 | 2.7 | 3.0 | 3.7 |
| May | -110 | -10.0 | -16.1 | -25.2 | -13.2 | -13.3 | -15.2 | -13.2 | -13.5 | -16.8 |
| June | -85 | -5.0 | -4.3 | -12.7 | -6.3 | -5.8 | -6.0 | -5.7 | -6.0 | -7.3 |
| **Spring** | **-51** | **-4.2** | **-5.9** | **-10.5** | **-5.5** | **-5.4** | **-6.2** | **-5.5** | **-5.6** | **-6.9** |
| July | -120 | -8.1 | -8.4 | -21.0 | -11.0 | -10.7 | -11.3 | -10.6 | -11.0 | -13.5 |
| August | -180 | -10.3 | -7.4 | -27.4 | -15.8 | -14.7 | -15.2 | -14.8 | -15.2 | -18.7 |
| September | -140 | -8.7 | -9.0 | -22.7 | -13.3 | -13.1 | -13.7 | -13.0 | -13.3 | -16.7 |
| **Summer** | **-145** | **-9.0** | **-8.3** | **-23.7** | **-13.4** | **-12.8** | **-13.4** | **-12.8** | **-13.2** | **-16.3** |
| October | 10 | 0.6 | 0.6 | 1.6 | 1.0 | 0.9 | 1.0 | 1.0 | 1.0 | 1.3 |
| November | 450 | 35.0 | 42.3 | 93.3 | 59.7 | 59.1 | 62.7 | 59.7 | 63.0 | 78.0 |
| December | 590 | 51.6 | 80.6 | 130.0 | 82.6 | 83.9 | 93.2 | 84.8 | 92.6 | 114.5 |
| **Fall** | **350** | **29.0** | **41.2** | **74.8** | **47.6** | **47.9** | **52.2** | **48.4** | **52.1** | **64.5** |
| January | 540 | 40.0 | 49.7 | 99.7 | 64.8 | 60.9 | 66.5 | 60.6 | 65.5 | 81.0 |
| February | 500 | 49.3 | 84.8 | 123.4 | 71.4 | 72.5 | 82.1 | 72.8 | 81.0 | 100.0 |
| March | 385 | 40.3 | 56.1 | 102.3 | 61.3 | 60.03 | 65.5 | 59.4 | 63.2 | 78.1 |
| **Winter** | **475** | **43.1** | **63.1** | **108.1** | **65.7** | **64.5** | **71.1** | **64.1** | **69.7** | **86.0** |
| **Annual** | **157** | **14.7** | **22.5** | **37.1** | **23.6** | **23.4** | **25.9** | **23.5** | **25.7** | **31.8** |
| | | | | | | | | | | |
| **Annual gCm$^{-2}$yr$^{-1}$** | - | 64.6 | 98.8 | 163.1 | 103.6 | 102.8 | 113.9 | 103.3 | 113.0 | 139.7 |
| **Deviation from *SD20-fit*** | - | -54 % | -29 % | +17 % | -26 % | -26 % | -18 % | -26 % | -19 % | - |




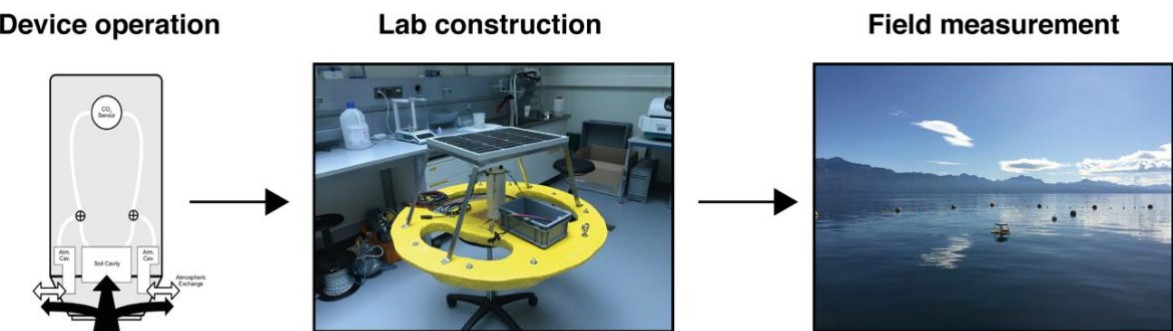

**Figure A1: Schematics of eosFD operation (eosense.com) followed by its mini-platform construction and its positioning**
**for measurements in the field (Lake Geneva at LéXPLORE platform). The raft design also complies with recommendations to minimize artificial turbulence induced by the chamber's walls, with 10 cm long-edges entering the water (Vachon et al., 2010).**

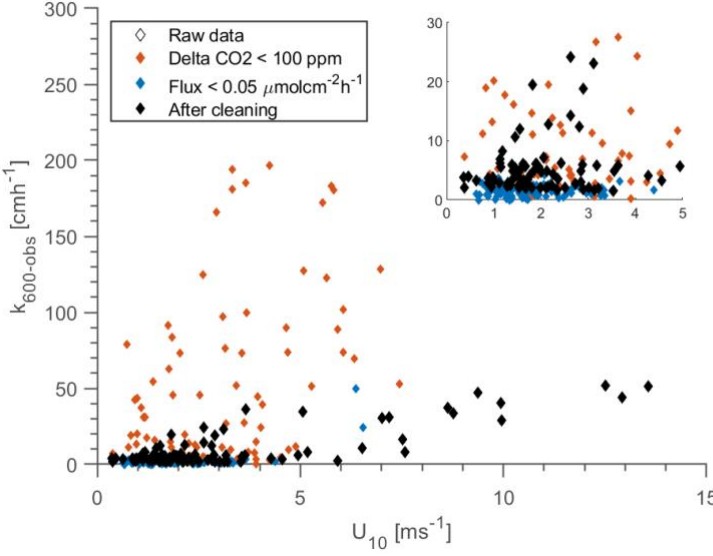


**Figure A2: Visualisation of 304 observed $k_{600}$ during the five periods of flux measurements (i.e., 13th–14th June 2019, 27th–28th August 2019, 1st–5th October 2019, 18th–20th December 2019, and 20th–26th February 2020).**


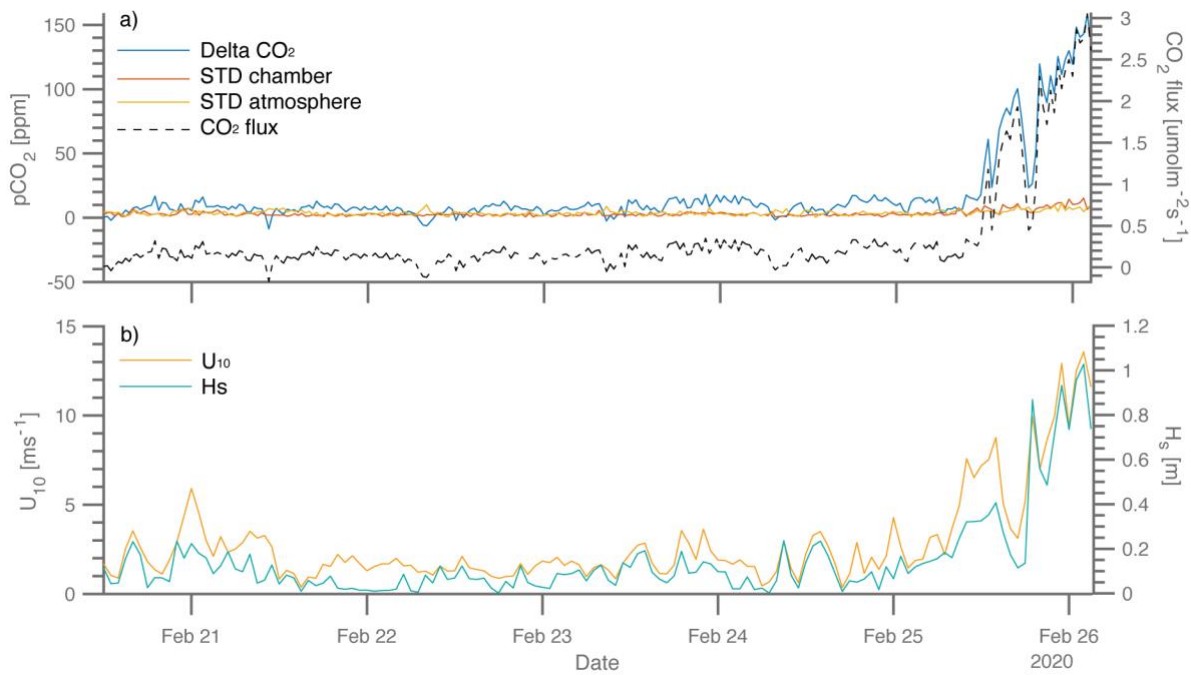

**Figure A3: a) Raw outputs of the eosFD during one period of $CO_2$ flux measurements; $\Delta CO_2$ between both cavities of**
**measures (atmosphere cavity and chamber cavity) (blue line); Standard deviation of each cavity between two**
**automated flushing (30 minutes of interval), Chamber cavity (red line), Atmosphere cavity (yellow line); and the $CO_2$**
**flux (black dash line). b) Temporal evolution of $U_{10}$ and $H_s$ during the same period than $CO_2$ flux measurements.**
**Increase in flux on 25th February corresponding to increase in wind speed and waves.**


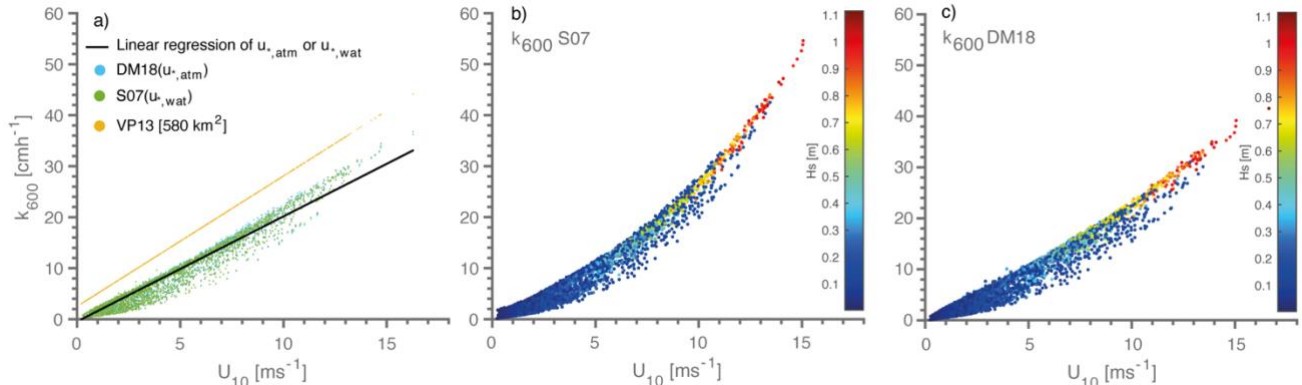

**Figure B1: a) Comparison of Soloviev et al. (2007) and Deike and Melville (2018) for the first order function of friction**
**velocity at the water side ($u_{*,wat}$) (blue points) and at the atmosphere side ($u_{*,atm}$) (green points) with their linear**
**regression (black line); the linear function of Vachon and Prairie (2013) for a lake size of 582 km² (yellow points) as**
**well as the linear regression from ; b) Visualisation of *S07* with empirical parameterization of bubble term (Woolf,**
**1997) regardless of wave height in function of wind speed at 10 m; c) Visualisation of *DM18* in function of wind speed,**
**only effect of bubble term from 10 ms⁻¹.**



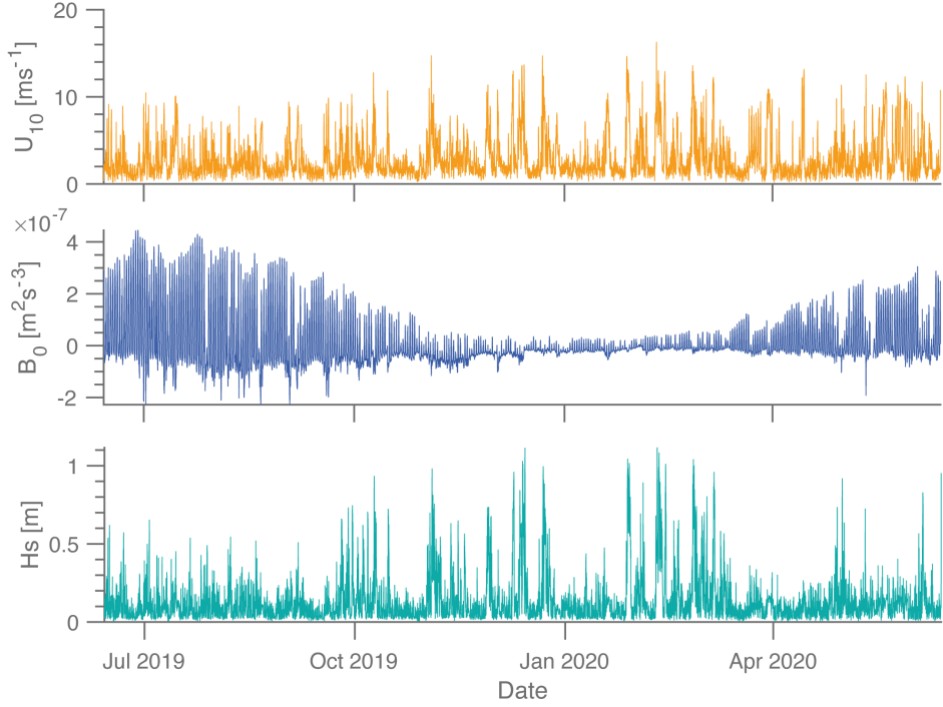

**Figure C1: Annual evolution of 3 main inputs of *k*-models; Wind speed at 10 m ($U_{10}$); Buoyancy flux at surface ($B_0$); Significant wave height ($H_s$).**