# Peer review of "Accounting for surface waves improves gas flux estimation at high wind speed in a large lake"

_Earth System Dynamics, 2021_

## Author Comment (AC1)

**Response to Anonymous Referee #1**

We thank you very much for your constructive comments and suggestions.
Below the reviews are reproduced in black font, our replies are interspersed in **blue** while preliminary updates of the text are in ***green***.

In this well-constructed study, Perolo and colleagues evaluated the performance of various empirical and process-based gas exchange (k) models using a combination of high- frequency k and weather measurements, intending to find the best model to accommodate high wind and high wave heights conditions often found in large lakes such as Lake Geneva. To be able to account for the major processes affecting k on such a large lake (i.e. wind shear, buoyance flux and wave motion including bubble enhancement), the authors cleverly combined and adapted existing process-based models first developed for the ocean. The chosen adapted model was proven the most accurate and flexible to a wide range of wind speeds and wave heights.

Investigating k at high wind and wave heights conditions is an important and overlooked aspect of k dynamics in the context of episodic events. The addition of waves and bubble enhancement to lakes k models is novel, to my knowledge. I particularly liked the cumulative k analysis as it clearly demonstrates the disproportionally important role of rare periods of high winds and waves. It also elegantly shows how each model responds as a whole to the distributions of wind and wave that actually occurs on a large lake. I also think that Figure 1 is very useful and accurately summarizes the main processes and the predictive models with their respective variables used.

Overall, I found the manuscript well written, well organized and easy to follow. The references are appropriate and the methods clear. I only have a few general and specific comments that could potentially improve the manuscript.

Reply: We very much appreciate this overall positive assessment.

**General comments:**

1. The lack of spatial integration (due to only one measurement station) is only discussed briefly in the conclusion. Wind and waves fields are different in other parts of the lake and this can have a large impact on the conclusions made regarding CO2 fluxes. I suggest the authors expand this part and move to the discussion section.

Reply: Following your comment and those of the second reviewer, we developed this aspect in the discussion, by computing k-values over the lake surface using the COSMO wind-model of Meteoswiss for two representative's dates with windspeeds > 10 m s$^{-1}$. Since this is mostly a model-based illustration of the spatial implications of k-models, this section was added to the discussion. Please find below the section added, after section 4.2.2.

[revised manuscript text omitted]

Finally, we will clarify in section 4.2.2 that our estimate of CO2 fluxes remains a coarse estimate with main goal to scale the effects of wave integration on annual CO2 fluxes. More data would be needed to provide an accurate estimate of CO2 emissions at the lake scale.

2. In undersaturated CO2 conditions (high pH), there is the possibility at CO2 fluxes are enhanced chemically (aka chemical enhancement factor). This usually happens during productive periods (summer with undersaturated pCO2), where CO2 is rapidly consumed chemically (hydration) at the very surface of water (Wanninkhof and Knox, 1996), enhancing the CO2 influx from the atmosphere, but not affecting pCO2 measurements at a deeper depth. If this is the case (at about pH > 8), it would result in an overestimation of observed k values measured from pCO2 and flux chambers, especially under calm wind conditions. In the manuscript, the chemical enhancement factor was not taken into account, and I think this should be justified. I do not think this would change the main conclusions of the paper, but it may potentially slightly affect the parameterization (SD20-fit) and the models evaluations.

Reply: We agree that the notion of chemical enhancement, which is likely to occur in Lake Geneva in the summer period, should be specified. Chemical enhancement is not relevant for the time-periods on which we tested the different k-models.

Indeed, our analysis of k-models accuracy is based on data recorded during the periods of oversaturation (December and February, i.e.: when pH data are well < 8) since summer data did not pass the quality control (CO2 flux too low or delta CO2 too low). Consequently, the parameterization of the SD20-fit (now SD21-fit) model should not be impacted by the absence of integration of the chemical enhancement factor. Besides, our parametrization mainly impacts the gas exchange velocity results for wind > 5 ms$^{-1}$ when chemical enhancement was not expected to have a significant effect (see for instance Figure 2 in Wanninkhof and Knox, 1996).

Following this comment, we added a few sentences explaining these reasons in section 2.3 (line 161):

"Noteworthily, the chemical enhancement factor (Wanninhof and Knox, 1996) was not considered in this equation since the fluxes retained corresponded to conditions of moderate pH (i.e., < 8) where such a process should not affect calculations."

However, we agree chemical enhancement should be accounted for summer and annual estimates of k-values, such as on their consequences on estimated CO2 fluxes (Table 3). If this factor is to be taken into consideration at high pH, it should be incorporated into any k-models whether wind- or process-based.

As a reminder, the formulation of Wanninkhof and Knox (1996) shows that the chemical enhancement factor $\alpha$ depends on the stagnant boundary layer thickness z following:

$$\alpha = T / ((T - 1) + \tanh(Q \cdot z)/(Q \cdot z)) \tag{10}$$

While z itself depends on k as follows:

$$z = D/k \tag{1}$$

Therefore, estimates for chemical enhancement would themselves depend on the chosen k-models, with even further consequences on estimated CO2 fluxes.

We therefore computed estimated fluxes with and without accounting for the chemical enhancement and completed Table 3 and section 4.2.2 as below:

[revised manuscript text omitted]

3.  Significant wave heights (Hs) were not measured but predicted from U10 and fetch. Do the authors have any idea of uncertainties associated with the predicted Hs?

Reply: Similar comment was also raised by reviewer 2. We are unfortunately unable to provide a quantification of the uncertainties from field observations.

Therefore, we rephrased this section from the methodology concerning Hs and its definition (line 132-136):

"This variable $H_s$ is defined as the average height of the highest one-third of the waves (crest to trough) corresponding to the thickness over which the wind can push laterally (Wüest and Lorke, 2003). This equation is equivalent to the formulation by Carter (1982) that is more widely used in the oceanic literature. Simon (1997) tested the model for significant wave heights in Lake Neuchâtel (a lake close to Lake Geneva) with a fetch distance of 9 km. These results showed that the significant wave height in this lake was consistent with this oceanic formulation. However, Simon (1997) highlighted that the Joint North Sea Wave Project (JONSWAP) wave breaking parametrization did not hold for winds greater than 5 m s$^{-1}$ producing faster wave breaking and with a

higher probability in the case of not fully developed surface waves. Such lake waves are characterized by steeper slopes that favour their wave breaking and wave action (Wüest and Lorke, 2003)."

We instead stress that such measurements are needed in large lakes to better constrain the air-water exchanges under various surface roughness in the new section (4.2. see below)

4. The terms *SD20* and *S20* (in Fig. 5) are presented in the same way as the other published models (i.e. first letter of author name and the year of publication). In this case, does SD stands for Soloviev and *D* for Deike? and why 20? Please explain in the text and/or in the figure caption.

Reply: We added an explanation for the name given to the model on line 251 at the end of the methodology section.

"Our adapted model for lake includes a refined parametrization of the wave action term $\varepsilon_w$ from *S07* along with the bubble term from *DM18*. For these reasons, the model will be called *SD21* in the rest of the manuscript. In addition, for the appellation *SD21-fit*, the $a_1$ parameter of Eq. (2) and the $A_B$ parameter of Eq. (15) were fitted to the $k_{600}$ observations ($a_1 = 0.33$ and $A_B = 3 \ 10^{-5}$ m$^{-2}$ s$^2$)."

The 20 there referred to the year when this article was initially written. We will transform the name to *SD21* throughout the text and figures.

Specific comments:

L42: I think it should be Cole and Caraco (1998), as it is commonly called, instead of Cole et al.

Reply: We agree. We made a reference mistake. It will be changed for Cole and Caraco.

*"... starting with the Cole and Caraco (1998) seminal paper..."*

L138: As this is a novel method, I think more details on the automated (forced diffusion) flux chamber should be given. How this chamber is different from the more traditional floating chamber?

Reply: Following your comment and that of the second referee, we added a few lines (after line 142) to the flux measurement methodology and created a new figure for the appendix to show the different designs.

"One typical problem with floating chambers arises from the possible atmospheric leakage under rough surface (Fig. A2a). To work around this problem, Vachon et al. (2010) advise to create 10 cm long-edges entering the water (Fig. A2b) and this design also reduces artificial turbulence generated by the chamber's walls at surface. A second typical issue with this method is potential flux enhancement by artificial (chamber-generated) turbulence. This was also studied in Vachon et al. (2010), who demonstrated that the overestimations by this effect can be as high as 1000 % at low wind but less than 50 % when the wind speed exceeds 4 m s$^{-1}$ in large lakes. At even higher wind speed, this overestimation should further decrease because the surface water turbulence becomes much greater than that produced by the floating chamber. Thereby, our flux chamber was specifically conceived to increase stability under calm and windy conditions and limiting artificial turbulence, but we do not exclude a bias at low and moderate wind (Fig. A1, Fig. A2c).

Regarding the operation of the eosFD, it has two independent cavities: one for the chamber and one for the atmosphere (Fig. A1). These are connected to the same $CO_2$ sensor by a pump which sends at regular intervals (about 20 s) either the chamber gas or the air gas to the sensor and then completely flushes the chamber cavity according to the programmed measurement timestep (15-minutes or 30-minutes). The advantage of this new instrument is therefore to have a constant monitoring of the chamber's variation, but also of the atmosphere. In addition, the use of the same $CO_2$ sensor for the two measurements limits the need for intercalibration between $CO_2$ sensors. We tested the performance of the floating chamber by comparing the standard deviation of the $CO_2$ concentrations of the atmosphere and in the chamber estimated from two separated cavities (Fig. A1; Risk et al., 2011). We did not observe any difference in the standard deviation between high and low wind conditions (Fig A3), suggesting that the measured fluxes remained reliable at high wind speed without leakage of the chamber."

[Figure]

**Figure A2new: a) Classic floating chamber; b) Floating chamber with 10 cm long-edges; c) Platform design used in this study: 10 cm long-edges, rounded-edges, and flat and long water wings.**

L235: I don't know what the "^" symbol means here?

Reply: We will remove this typo.

L280. Missing dot.

Reply: We will correct this typo.

L362: But specific calibration (a and A) would also be needed in process-based models like it is the case for Lake Geneva.

Reply: We agree with this remark. The process-based models can also have a recalibration with these two empirical parameters ($a_1$ and $A_b$) in different systems. However, they remain more consistent than wind-based models even without refining these parameters. Also, we had to recalibrate, as they had never been implemented in lakes with the surface wave effect included. Therefore, we will remove this sentence.

Fig. C1. The models used only negative buoyancy flux, which induces turbulence by convection. I wonder what is the effect of positive buoyancy during heating on k. Could it reduce the effect of wind shear as it suppresses turbulence?

Reply: This is an important point that we now discuss in the manuscript. We now clearly state that our model only considers buoyancy flux when this terms directly enter into the turbulent kinetic equation as production term. We acknowledge that a positive buoyancy flux should reduce the effect of wind shear turbulence production to a yet to be established extent that would depend on the near surface stratification. Assuming that typically 20% of the production term is transferred to mixing (Ivey et al. 2008), a first order approximation would be to reduce the effect of the wind shear turbulence production by a similar amount in our equation in the case of net heating of the lake. Another important aspect is the role of stratification that offset the measured CO2 values with respect to those at the interface. This subject deserves an attention beyond the scope and data availability of this work.

We added this sentence in section 2.4.2 (line 231):

"A second source of dissipation at the surface is the convection ($\varepsilon_c$) resulting from surface cooling. In the SRM formulation, only the negative buoyancy flux is considered when this term directly enters into the turbulent kinetic equation as production term. The combination of wind shear and free convection near a boundary is described by the Monin-Obukhov similarity theory (MOST) with a general form derived from a turbulent kinetic energy balance (Lombardo and Gregg, 1989; Tedford et al. 2014):"

Change of this reference

Simon A.: *Turbulent mixing in the surface boundary layer of lakes.* PhD thesis no. 12,272. Swiss Fed. Inst. Technol. (ETH), Zurich, 1997.

Add references

Ivey, G.N., Winters, K.B., and Koseff, J.R.: Density stratification, turbulence, but how much mixing? Annu. Rev. Fluid Mech., https//doi.org/10.1146/annurev.fluid.39.050905.110314, 2008.

Other changes

We add in table 1 (CC98 for Calibrated range) Area (0.15-490 km$^2$) because CC98 had taken other lake in the literature to perform his model.

We found a typo in the caption of the figure 6 with a sign error between 5b and 5c (greater and smaller). Here is the correction:

*Figure 6: a) Cumulative $k_{600}$ modelled over an annual cycle; b) Cumulative $k_{600}$ for wind > 5 m s$^{-1}$; c) Cumulative $k_{600}$ for wind ≤ 5 m s$^{-1}$.*

We also found small errors in the labels of y axes of figure 5 which will be corrected.

---

## Author Comment (AC2)

**Response to Anonymous Referee #2**

We thank you very much for your constructive comments and suggestions.
Below the reviews are reproduced in black font, our replies are interspersed in **blue** while preliminary updates of the text are in ***green***.

The authors analyze the enhancement of gas exchange velocities by surface waves in large lakes and explore the performance of a broad range of empirical and mechanistic models to predict this dependence based on wind speed and fetch length. As the authors point out correctly, the effect of surface waves on gas exchange is neglected in most studies, and observations are largely lacking – particularly in lakes. By analyzing CO2 flux measurements obtained during different seasons in Lake Geneva, the authors demonstrate, that waves have a potentially significant influence on gas exchange velocity and gas fluxes in large lakes, although sufficiently large waves occur during rare events only.

The manuscript addresses an important research gap and makes an original contribution to advancing the prediction of gas exchange in numerical models. It is well written and organized.

Reply: We appreciate this overall positive assessment and believe we can address the Reviewer's concerns as detailed below.

The largest shortcoming of the study is certainly the lack of wave observations (wave height was derived from wind speed using a model adopted from marine systems). Apparently there are only very few wave measurements from lakes available. I suggest that the authors emphasize this issue in their discussion and mention the need for direct wave observations in lakes in future studies. As listed below, I have a few additional comments, which can be addressed in minor revisions of the manuscript.

Reply: This is indeed an important point that was also raised by Reviewer 1. The lack of direct wave observations is a major shortcoming of our study. We put more attention on this in the discussion where we also developed a small part on spatial integration as proposed by referee 1 (see below reply to comment L. 173). We detailed the different methods to quantify these wave fields (see below reply to comment L. 173). As a matter of fact, direct wave measurements are one of the upcoming challenges to be undertaken at the Lexplore platform.

We stressed the need for more observations of surface waves in lakes. However, our study is still solid as based on one of the few studies of surface waves in lakes conducted in a similar lake (Lake Neuchatel, another deep lake located a few kilometers away from Lake Geneva). More details on the changes made are detailed below (L. 135 and L. 173).

**Detailed comments:**

1. L. 31: "...these approaches suffer from limited time and space integration..." I don't think that this applies to EC measurements.

Reply: We will rephrase this sentence like this and add a new reference:

*"... CO₂ fluxes can be directly measured with floating chamber or eddy covariance systems (Vachon et al., 2010; Vesala et al., 2006). However, both approaches have their own constraints. The former suffers from limited time and space integration (from minutes to hours, and centimetres to metres; Klaus and Vachon, 2020), while the latter remains technically difficult and can be influenced by non-local processes (entrainment from the shore or advection; Vachon et al., 2010; Ester et al., 2020)."*

2. L. 52: "with order larger than unity" consider rewording

Reply: We rephrased this sentence like this:

3. L. 135: I suggest being more clear here: the wind observations of Simon showed that the JONSWAP parameterization did not hold for wind speed > 5 m/s

Reply: We have rewritten the passage of the methodology containing the explanations of the formula for significant wave heights (line 132-136) to clarify for the reader.

"This variable $H_s$ is defined as the average height of the highest one-third of the waves (crest to trough) corresponding to the thickness over which the wind can push laterally (Wüest and Lorke, 2003). This equation is equivalent to the formulation by Carter (1982) that is more widely used in the oceanic literature. Simon (1997) tested the model for significant wave heights in Lake Neuchâtel (a lake close to Lake Geneva) with a fetch distance of 9 km. These results showed that the significant wave height in this lake was consistent with this oceanic formulation. However, Simon (1997) highlighted that the Joint North Sea Wave Project (JONSWAP) wave breaking parametrization did not hold for winds greater than 5 m s$^{-1}$ producing faster wave breaking and with a higher probability in the case of not fully developed surface waves. Such lake waves are characterized by steeper slopes that favour their wave breaking and wave action (Wüest and Lorke, 2003)."

We instead stress that such measurements are needed in large lakes to better constrain the air-water exchanges under various surface roughness in the new section (4.2. see below)

4. L. 144-47: I cannot really follow the argumentation using the standard deviation. Maybe this needs to be explained in a better way. Besides leakage of the chamber in a wavy environment, there can also be flux enhancement by artificial (chamber-generated) turbulence.

Reply: Following your comment and that of the first referee, we added a few lines (after line 142) to the flux measurement methodology and created a new figure for the appendix to show the different designs.

"One typical problem with floating chambers arises from the possible atmospheric leakage under rough surface (Fig. A2a). To work around this problem, Vachon et al. (2010) advise to create 10 cm long-edges entering the water (Fig. A2b) and this design also reduces artificial turbulence generated by the chamber's walls at surface. A second typical issue with this method is potential flux enhancement by artificial (chamber-generated) turbulence. This was also studied in Vachon et al. (2010), who demonstrated that the overestimations by this effect can be as high as 1000 % at low wind but less than 50 % when the wind speed exceeds 4 m s$^{-1}$ in large lakes. At even higher wind speed, this overestimation should further decrease because the surface water turbulence becomes much greater than that produced by the floating chamber. Thereby, our flux chamber was specifically conceived to increase stability under calm and windy conditions and limiting artificial turbulence, but we do not exclude a bias at low and moderate wind (Fig. A1, Fig. A2c).

Regarding the operation of the eosFD, it has two independent cavities: one for the chamber and one for the atmosphere (Fig. A1). These are connected to the same $CO_2$ sensor by a pump which sends at regular intervals (about 20 s) either the chamber gas or the air gas to the sensor and then completely flushes the chamber cavity according to the programmed measurement timestep (15-minutes or 30-minutes). The advantage of this new instrument is therefore to have a constant monitoring of the chamber's variation, but also of the atmosphere. In addition, the use of the same $CO_2$ sensor for the two measurements limits the need for intercalibration between $CO_2$ sensors. We tested the performance of the floating chamber by comparing the standard deviation of the $CO_2$ concentrations of the atmosphere and in the chamber estimated from two separated cavities (Fig. A1; Risk et al., 2011). We did not observe any difference in the standard deviation between high and low wind conditions (Fig A3), suggesting that the measured fluxes remained reliable at high wind speed without leakage of the chamber."

[Figure]

**Figure A2new: a) Classic floating chamber; b) Floating chamber with 10 cm long-edges; c) Platform design used in this study: 10 cm long-edges, rounded-edges, and flat and long water wings.**

5. L. 173: there is a square missing in the equation for surface shear stress

Reply: We corrected this typo and added the square missing in this equation.

6. L. 334 (and elsewhere): I'm not sure if the cumulative k is a very illustrative quantity (as the numbers are kind of meaningless). Did you consider analyzing the cumulative mean values of k instead? (cumulative sum normalized by number of observations)

Reply: We understand your concern about the significance of the "cumulative k" but we prefer to keep it for the following reason. The cumulative k was used to highlight temporally the contribution of the wind on k. With this, we see that strong wind (> 5 m s-1) accounted for ~40% of the cumulative k while representing less than 15% of the time.

7. L. 171: missing word

Reply: We are not sure what you think about the missing word. Do you mean this?

*"… 2.4.1. Wind shear stress…"*

8. L. 173 (and elsewhere): when discussing the frequency of occurrence of waves exceeding a certain height and corresponding enhancement of k and fluxes, it is important to keep in mind that these estimated are site-specific (within the lake). I suggest that the authors briefly discuss to what extent the observations made at the platform are representative for the entire lake. Given the distribution of wind directions and lake geometry – are there sites where wave can be expected to make large/smaller contributions?

Reply: Following your comment and those of the first reviewer, we suggest implementing in the discussion a section on the spatial variability of wind-waves after the section 4.2.2.

[revised manuscript text omitted]

Finally, we will clarify in section 4.2.2 that our estimate of CO2 fluxes remains a coarse estimate which main goal is to scale the effects of wave integration on annual CO2 fluxes. More data would be needed to provide an accurate estimate of CO2 emissions at the lake scale.

9.  L. 407 ff.: how well do the monthly mean pCO2 values represent the conditions during high wind speed (high waves)? As pointed out later, pCO2 could be expected to be much higher

(entrainment) or much lower (depletion of CO2 in the surface layer due to strong outgassing) during these events. Do the authors have observations from such events?

Reply: We agree that annual flux estimates will gain in accuracy using high frequency data for surface CO2. However, the purpose of this paper (section 4.2.2) is to compare the fluxes estimated through different k models, in order to scale the importance of integrating/omitting surface waves at an annual scale. For this aim, monthly data for surface CO2 are sufficient to quantify the effects of waves on gas exchange velocities. Using high-frequency surface CO2, we would need to tease apart wind-wave effect on CO2 fluxes due to changes in k only from those due changes in surface CO2 following wind-driven mixing or internal waves, and we fear this would finally dilute the focus of the paper. We will make the purpose of the exercise clearer for the reader in this part and highlight the need to obtain high frequency data of CO2 in water to study these two possibilities (entrainment - depletion) when it comes to accurate estimates of CO2 fluxes at the lake-water interface.

10. I suggest to add the observed fluxes to Table 3 to allow others to use or to reproduce the results presented here.

Reply: It is difficult to integrate the observed fluxes in Table 3, because they are not in adequacy with the time steps of the table and were recorded only during 5 different months. Moreover, as explained above, this table allows to compare the results of the k models in terms of flux and do not have to make an overall estimate of the lake knowing the temporal and spatial variability of the CO2 concentration at the surface. However, measured flux data (chamber), as well as the corresponding CO2 concentrations (sensors) will be made available on Zenodo upon acceptance so others could use and reproduce the results (see data availability statement).

11. L. 442: "estimate an average fetch value depending on the wind direction and the geometry of the lake" But the dependence of k on fetch is non-linear (fetch^1/3?). Should the spatial averaging take this into account?

Reply: You are right. The dependence between k and fetch is non-linear. This approach is the simplest to integrate a fetch in the calculations. However, it could propagate errors spatially. We will therefore apply a spatial average fetch in the new part of the discussion on spatial integration as showed above (Figure 8g, diamonds and cross).

Added reference

Ester, L., Rutgersson, A., Nilsson, E. and Sahlée, E.: Non-local impacts on eddy-covariance air-lake $CO_2$ fluxes, Boundary-Layer Meteorol., 178, 283–300, https://doi.org/10.1007/s10546-020-00565-2, 2021.

Other changes

We add in table 1 (CC98 for Calibrated range) Area (0.15-490 km$^2$) because CC98 had taken other lakes in the literature to perform his model.

We found a typo in the caption of the figure 6 with a sign error between 5b and 5c (greater and smaller). Here is the correction:

Figure 6: a) Cumulative $k_{600}$ modelled over an annual cycle; b) Cumulative $k_{600}$ for wind > 5 m s$^{-1}$; c) Cumulative $k_{600}$ for wind $\leq$ 5 m s$^{-1}$.

We also found small errors in the labels of y axes of figure 5 which will be corrected.